# Neuronal glutamate transporters control reciprocal inhibition and gain modulation in D1 medium spiny neurons

**Maurice A Petroccione, Lianna Y D'Brant, Nurat Affinnih, Patrick H Wehrle, Gabrielle C Todd, Shergil Zahid, Haley E Chesbro, Ian L Tschang, Annalisa Scimemi\***

SUNY Albany, Department of Biology, Albany, United States

**Abstract** Understanding the function of glutamate transporters has broad implications for explaining how neurons integrate information and relay it through complex neuronal circuits. Most of what is currently known about glutamate transporters, specifically their ability to maintain glutamate homeostasis and limit glutamate diffusion away from the synaptic cleft, is based on studies of *glial* glutamate transporters. By contrast, little is known about the functional implications of *neuronal* glutamate transporters. The neuronal glutamate transporter EAAC1 is widely expressed throughout the brain, particularly in the striatum, the primary input nucleus of the basal ganglia, a region implicated with movement execution and reward. Here, we show that EAAC1 limits synaptic excitation onto a population of striatal medium spiny neurons identified for their expression of D1 dopamine receptors (D1-MSNs). In these cells, EAAC1 also contributes to strengthen lateral inhibition from other D1-MSNs. Together, these effects contribute to reduce the gain of the input-output relationship and increase the offset at increasing levels of synaptic inhibition in D1-MSNs. By reducing the sensitivity and dynamic range of action potential firing in D1-MSNs, EAAC1 limits the propensity of mice to rapidly switch between behaviors associated with different reward probabilities. Together, these findings shed light on some important molecular and cellular mechanisms implicated with behavior flexibility in mice.

**\*For correspondence:**
scimemia@gmail.com

**Competing interest:** The authors declare that no competing interests exist.

## Editor's evaluation

This fundamental study outlines the role of the neuronal glutamate transporter (EAAC1) in striatal function and behavior. The evidence supporting the conclusions is compelling, with rigorous studies spanning genetic approaches, physiology, and behavior. This work will be of general interest to those studying striatal biology as well as the neural control of behavior.

## Introduction

The *neuronal* glutamate transporter EAAC1, encoded by the *Slc1a1* gene, is distributed broadly throughout the brain (*Rothstein et al., 1994*; *Shashidharan et al., 1997*), with plasma membrane surface density values thought to be significantly lower than those of *astrocytic* glutamate transporters (*Holmseth et al., 2012*). Glutamate binding to glutamate transporters activates currents that can be easily recorded in heterologous expression systems and astrocytes, where the density of expression of these molecules is high (*Wadiche et al., 1995a*). In most neurons, recording glutamate transporter mediated currents continues to be elusive (*Holmseth et al., 2012*). This finding sparked some doubts on the functional relevance of neuronal glutamate transporters like EAAC1 in the brain (*Holmseth et al., 2012*). Despite this concern, multiple lines of evidence point to the fact that EAAC1 is an

important player in the regulation of synaptic function and behavior. For example, in the hippocampus, EAAC1 limits NMDA receptor activation and increases GABA release (*Diamond, 2001*; *Scimemi et al., 2009*; *Mathews and Diamond, 2003*). In the dorsolateral striatum (DLS), EAAC1 limits activation of group I metabotropic glutamate receptors (mGluRI) and is associated with increased execution of stereotyped motor behaviors (*Bellini et al., 2018*). While most of these works relied on the use of *Slc1a1⁻/⁻* mice (*Peghini et al., 1997*), other studies that used overexpression models of EAAC1 showed behavioral abnormalities, including increased anxiety-like behaviors (*Delgado-Acevedo et al., 2019*; *Escobar et al., 2021*) and reduced responses to amphetamine-induced hyperlocomotion (*Zike et al., 2017*). The emerging picture is that maintaining optimal levels of EAAC1 expression is key for proper execution of a wide range of complex motor behaviors and habitual actions, many of which are critically dependent on the activity of the DLS (*Gremel and Costa, 2013*; *Yin et al., 2004*; *Yin et al., 2006*).

What continues to puzzle the field is how we can reconcile the obvious behavioral abnormalities of *Slc1a1⁻/⁻* mice with the apparent inability to record EAAC1-mediated currents in neurons. A possible way out of this conundrum comes from considerations of the biophysical properties and sub-cellular distribution of this neuronal transporter. *First,* EAAC1 has a very low single channel conductance (~0.3 fS; *Grewer et al., 2000*) and is not evenly distributed along the plasma membrane (*Cheng et al., 2002*; *Conti et al., 1998*; *Rothstein et al., 1994*; *He et al., 2000*). *Second,* the fact that currents generated at a distance from the soma are dramatically reduced by electrotonic filtering and attenuation could make them particularly challenging to identify using somatic patch-clamp recordings (*Tønnesen et al., 2014*; *Svoboda et al., 1996*; *Rall, 1959*). *Third,* tentative estimates of the *average* surface density of expression EAAC1 obtained from immunoblot experiments do not provide information on the *local* density of expression of EAAC1 in dendritic spines and axonal boutons, where EAAC1 is thought to be confined. *Fourth,* the amplitude of local EAAC1-mediated currents does not provide a direct readout of the number of receptors that EAAC1 might protect from glutamate spillover (*Scimemi et al., 2009*). Together, these findings suggest that the inability to record EAAC1-mediated currents from the soma should not be interpreted as evidence of a lack of any functional role for glutamate uptake via EAAC1.

Interestingly, qualitative pre- and post-embedding ultrastructural works show that EAAC1 has a distinctive punctate peri-synaptic and post-synaptic expression in glutamatergic neurons, attributed to the presence of a specific domain in the C-terminal region that controls the targeting of EAAC1 to dendritic spines and shafts (*Cheng et al., 2002*; *He et al., 2000*). EAAC1 is also expressed in a population of GABAergic neurons identified by their expression of glutamic acid decarboxylase, the biosynthetic enzyme for GABA (*Conti et al., 1998*). In these neurons, EAAC1 shows a clustered expression in a subset of axonal presynaptic terminals (*He et al., 2000*; *Rothstein et al., 1994*), although these findings have been brought into question (*Holmseth et al., 2012*). In this case, the inconsistent results may be due to the limited sensitivity and specificity of antibodies directed against EAAC1 used for immunolabeling studies, which is a historically challenging, if not insurmountable issue. Other labeling strategies based on the use of pharmacological tools have been hampered by the fact that there continues to be no drug that targets EAAC1 without also affecting other glial glutamate transporters (*Shimamoto et al., 1998*; *Tsukada et al., 2005*). For these reasons, functional studies combined with genetic approaches currently represent an ideal tool to fill current gaps of knowledge on the functional roles of EAAC1 on synaptic communication in different brain areas.

Multiple lines of evidence suggest the existence of potential association between loss of function of EAAC1, altered synaptic activity in the striatum, and increased compulsive behaviors. *First,* the striatum is one of the brain regions with the most abundant expression of EAAC1. *Second,* hyperactivity of striatal circuits is associated with the emergence of compulsive behaviors (*Gassó et al., 2015*; *Gilbert et al., 2008*; *Menzies et al., 2008b*; *Abramowitz et al., 2009*; *Carmin et al., 2002*). *Third,* several genome-wide studies identified loss-of-function variants and single nucleotide polymorphisms of *SLC1A1*, the homolog gene encoding EAAC1 in humans diagnosed with OCD, autism spectrum disorder (ASD) and attention deficit/hyperactivity disorder (ADHD; *Veenstra-VanderWeele et al., 2012*; *Wendland et al., 2009*; *Porton et al., 2013*; *Stewart et al., 2007*; *Stewart et al., 2013a*; *Stewart et al., 2013b*; *Gadow et al., 2010*; *Brune et al., 2008*).

One of the hypotheses that has been brought forward to explain how EAAC1 may limit striatal hyperactivity is that impairing glutamate uptake via EAAC1 might lead to an increased ambient

glutamate concentration in the striatum, which in turn would trigger hyperactivity (**Porton et al., 2013**). This interpretation is difficult to reconcile with evidence that EAAC1 consitutes only ~5% of all glutamate transporters (**Holmseth et al., 2012**). Even if all EAAC1 molecules lost their ability to bind and transport glutamate, under steady-state conditions, the remaining ~95% of glutamate transporters in glial cells would be able to keep the extracellular glutamate concentration at low nanomolar levels (**Herman and Jahr, 2007**; **Chiu and Jahr, 2017**). Accordingly, experimental data show that loss of EAAC1 does not change the extracellular glutamate concentration (**Rothstein et al., 1996**), a result supported also by our own previous works in the striatum (**Bellini et al., 2018**) and hippocampus (**Scimemi et al., 2009**). An alternative hypothesis comes from computational models of neural dynamics in cortico-striatal-thalamic networks, which suggest that loss of EAAC1 might contribute to striatal hyperactivity by changing the relative strength of synaptic excitation and inhibition (E/I) locally, in some or all medium spiny neurons (MSNs), the main long-range projection neurons in the striatum (**Rădulescu et al., 2017**). According to this model, local changes in E/I of striatal MSNs can alter the firing rates of neurons not only in the striatum but also in larger neural networks that include the cortex and thalamus, bringing them in a regime of hyperactivity (**Rădulescu et al., 2017**). Although these theoretical inferences provide potential explanations of how local changes in E/I may lead to changes in the activity of more complex neural networks and behaviors, an experimental investigation of how EAAC1 alters E/I in different populations of striatal MSNs has not been previously performed.

Here, we show that EAAC1 limits synaptic excitation and strengthens synaptic inhibition in the DLS (i.e., reduces E/I), by modulating the strength of excitatory synaptic transmission onto D1 dopamine receptor expressing MSNs (D1-MSNs), and reciprocal inhibition among D1-MSNs. Through these mechanisms, EAAC1 increases the offset of the input-output relationship of D1-MSNs, to levels that differ depending on the rate of incoming inhibition. Together, these findings indicate that impairing the activity of EAAC1 brings D1-MSNs in a hyperexcitable state where these cells fire more action potentials in response to (*i*) lower frequencies and (*ii*) smaller changes in the frequency of activation of thalamo-cortical excitatory inputs. The reduced lateral inhibition among D1-MSNs may promote task switching, increasing the likelihood that mice rapidly change from one motor behavior to another. These findings shed new light on the synaptic and circuit mechanisms through which EAAC1 modulates the activity of specific populations of MSNs, with potential implications for understanding the cellular basis of impulsivity and compulsive behaviors (**Grassi et al., 2015**; **Bari and Robbins, 2013**; **Boisseau et al., 2012**; **Benatti et al., 2014**; **Ettelt et al., 2007**).

## Results
### EAAC1 is expressed in DLS MSNs

Although there is evidence that EAAC1 is expressed in the DLS (**Holmseth et al., 2012**), it is not known whether its cellular distribution is limited to MSNs or differs between D1- and D2-MSNs. To address this, we performed an RNAscope FISH analysis in the DLS using probes for *Drd1a*, *Drd2* and *Slc1a1*, the genes encoding D1/2 dopamine receptors and EAAC1 in mice, respectively (**Figure 1A and B**). We identified a population of cells that did not express EAAC1, which we excluded from further analysis (24% of all cells stained with the nuclear marker DAPI; **Figure 1B**). The EAAC1-expressing cells could be further classified into three main groups using unsupervised clustering (**Figure 1C**), principal component analysis (**Figure 1D**) and a dimensionality reduction approach based on t-distributed stochastic neighbor embedding (t-SNE) analysis (**Figure 1F**). The graphs in **Figure 1C, D and F** describe the distribution of the raw data in each cluster, for different types of dimensionality spaces, and the fact that they all identify three data clusters indicate that this result is not dependent on the type of clustering approach being used. The three EAAC1-expressing cohorts of cells were identified as being either D1-MSNs (25%), D2-MSNs (24%; **Figure 1E**), or cells with a very low, but detectable expression of EAAC1 and D1/2 receptors (51%; **Figure 1E**). These findings suggest that: (*i*) in the DLS, EAAC1 mRNA is mostly expressed in MSNs compared to other types of cells, and (*ii*) at the mRNA level, the expression of EAAC1 is indistinguishable between D1- and D2-MSNs (**Figure 1G and H**).

### EAAC1 limits dendritic branching in D1-MSNs

There is ample evidence that the morphology of neurons is susceptible to changes in excitatory synaptic function and connectivity. This has been observed across multiple brain regions, including

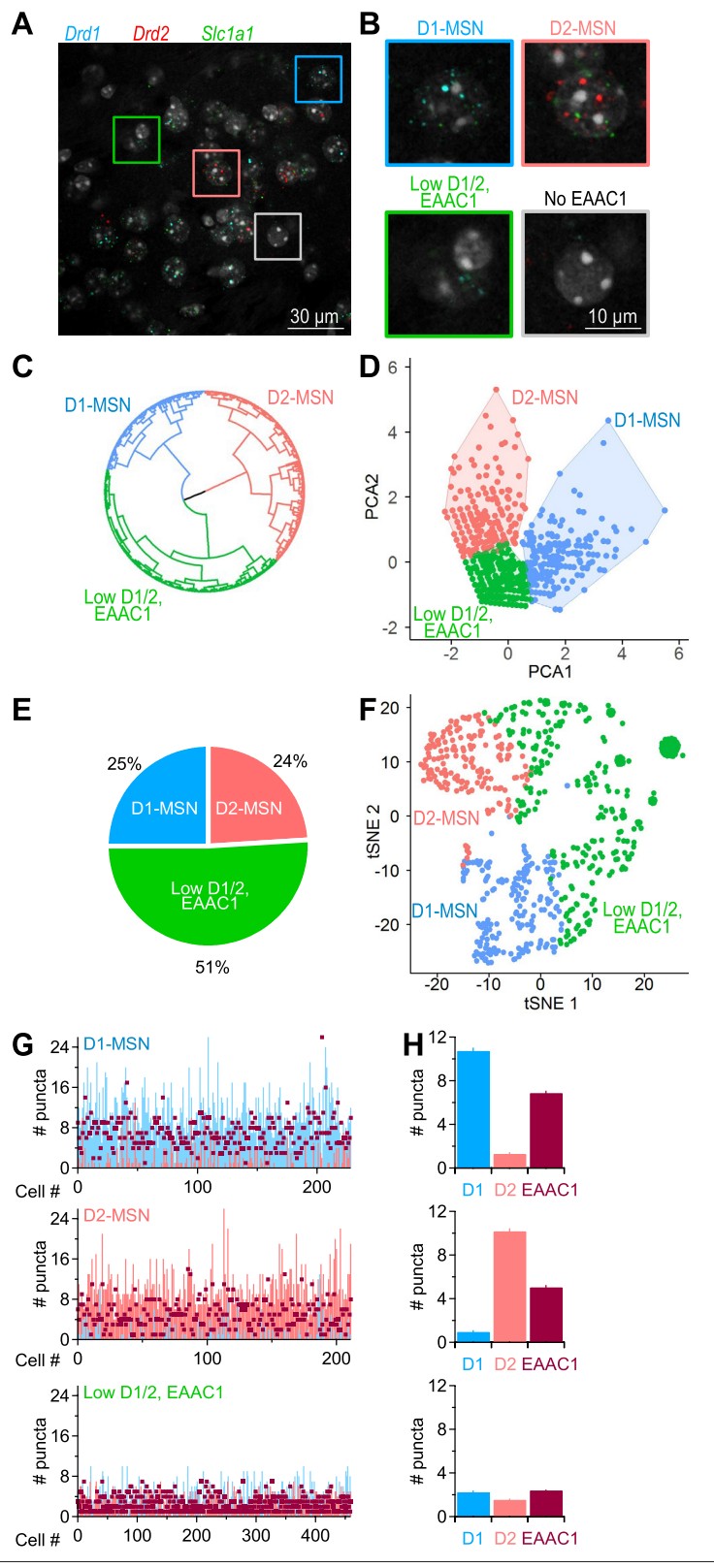

**Figure 1.** Distribution of EAAC1 mRNA in the DLS. (**A**) RNAscope FISH images of *Drd1* (*blue*), *Drd2* (*red*), and *Slc1a1* (*green*) mRNA expression in the DLS. These transcripts encode D1/2 receptors and EAAC1, respectively. (**B**) Annotated cell clusters identified with color-coded boxes in A, including cells expressing *Drd1* (i.e. D1-MSNs; *blue*), cells expressing *Drd2* (i.e. D2-MSNs; *salmon*), cells with low *Drd1/Drd2 /Slc1a1* EAAC1 expression (*green*),

*Figure 1 continued on next page*

*Figure 1 continued*

and cells with no detectable *Slc1a1* expression (which also had a very low expression of *Drd1/Drd2; gray*). (**C**) Circular dendrogram representation of RNAscope FISH data from EAAC1-expressing cells in the DLS. The dendrogram was obtained using the Ward's minimum variance method for hierarchical cluster analysis, using three as the number of groups cutting the tree (i.e., the optimal number of clusters obtained using gap statistics). The three main groups represent D1-MSNs (*blue*), D2-MSNs (*salmon*) and cells that express low levels of D1, D2 and EAAC1 (*green*). (**D**) Principal component analysis obtained using an enhanced k-means clustering with the three populations of cells. (**E**) Pie chart describing the percentage of the cell types identified with the hierarchical clustering. (**F**) Annotated clusters in the t-SNE map showing three specific groups of cells identified by the k-means algorithm based on their differential expression of D1, D2 and EAAC1. (**G**) Analysis of mRNA puncta in D1-MSNs (n=229 puncta; *blue*), D2-MSNs (n=212 puncta, *salmon*) and low D1/2, EAAC1 (n=460 puncta, *burgundy*). (**H**) Summary graph showing the mean number of puncta and S.E.M. measured in each of the three identified cell populations described in C.

---

the visual system, where the length and number of dendrites in stellate cells is reduced by visual deprivation, and the cerebellum, where Purkinje neurons have stunted dendrites when they receive fewer synapses from granule cells (*Coleman and Riesen, 1968*; *Wiesel and Hubel, 1963*; *Guillery, 1973*; *Friedlander et al., 1982*; *McAllister, 2000*; *Rakic, 1975*; *Rakic and Sidman, 1973*; *Sotelo, 1975*). Based on this, we reasoned that a structural analysis of MSNs could provide insights into potential functional changes in synaptic transmission associated with loss of EAAC1 expression. We reconstructed tdTomato-expressing and biocytin-filled MSNs from the DLS of mice that either expressed EAAC1 (i.e. *Drd1a*^Cre/+^:*Rosa26*^tm9/tm9^:*Slc1a1*^+/+^ and *Adora2a*^Cre/+^:*Rosa26*^tm9/tm9^:*Slc1a1*^+/+^ mice) or constitutively lacked it (i.e. *Drd1a*^Cre/+^:*Rosa26*^tm9/tm9^:*Slc1a1*^-/-^ and *Adora2a*^Cre/+^:*Rosa26*^tm9/tm9^:*Slc1a1*^-/-^ mice; *Figure 2A and B*). For simplicity, in the remainder of this manuscript, we collectively refer to all these mice as *Slc1a1*^+/+^ and *Slc1a1*^-/-^, respectively. A morphological Sholl analysis showed that D1-MSNs span a larger territory of the DLS neuropil and have a more intricate dendritic architecture in *Slc1a1*^-/-^ compared to *Slc1a1*^+/+^ mice (\*\*\*p=9.4e-6, r=0.90; *Figure 2C, left*). This effect that was not detected when comparing D2-MSNs in the two mouse strains (p=0.70, r=0.94; *Figure 2C, right*). Consistent with these findings, several other measures confirmed that the dendritic arbor of D1- but not D2-MSNs was larger in the absence of EAAC1. *First,* the maximum radius of the dendritic field of D1-MSNs was larger in *Slc1a1*^-/-^ mice (\*p=0.02), whereas that of D2-MSNs was similar in *Slc1a1*^+/+^ and *Slc1a1*^-/-^ mice (p=0.09; *Figure 2D, left*). *Second,* the center of mass of the dendritic arbor was located further away from the soma in D1-MSNs of *Slc1a1*^-/-^ compared to *Slc1a1*^+/+^ mice, whereas this was not the case for D2-MSNs (D1-MSN: \*\*p=7.6e-3; D2-MSN: p=0.17; *Figure 2D, center*). *Third,* the area of the neuropil covered by D1-MSNs was twofold larger in D1-MSNs of *Slc1a1*^-/-^ compared to *Slc1a1*^+/+^ mice (\*p=0.02), but it was similar in D2-MSNs (p=0.54; *Figure 2D, right*). *Fourth,* the total length of all dendrites, which receive synaptic inputs from other neurons, was ~twofold larger in D1-MSNs of *Slc1a1*^-/-^ compared to *Slc1a1*^+/+^ mice (\*p=0.01), but similar in D2-MSNs (p=0.27; *Figure 2E*). *Fifth,* the dendritic arbor of D1-MSNs in *Slc1a1*^-/-^ versus *Slc1a1*^+/+^ mice had more branches (\*p=0.03; *Figure 2F, left*) and branching points (\*p=0.03; *Figure 2F, center*), with a similar length of branch segments between branching points (p=0.70; *Figure 2F, right*). Dendrites of D2-MSNs had the same number of branches (p=0.18; *Figure 2F, left*), branching points (p=0.19; *Figure 2F, center*), and branch segment length (p=0.51; *Figure 2F, right*). These findings indicate that, through mechanisms that remain to be determined, EAAC1 is implicated with regulating the dendritic architecture of D1-MSNs.

## EAAC1 reduces spine number and size in D1-MSNs

Dendritic growth and an increased complexity of the dendritic arbor can be promoted by increased neuronal activity, in vitro and in vivo (*Yu and Malenka, 2003*; *Redmond et al., 2002*; *Sin et al., 2002*). Since changes in dendritic morphology are often concurrent with changes in synaptic structure and function, we asked whether the structural changes in the dendritic arborization of D1-MSNs in the absence of EAAC1 might be associated with changes in the number and functional properties of excitatory synaptic inputs onto these cells (*Peng et al., 2009*). We addressed this question by analyzing the density and subtype distribution of dendritic spines, the anatomical correlates of excitatory synapses, in biocytin-filled MSNs. Spine size scales with AMPA receptor content and the release probability of the presynaptic terminal with which they are in contact (*Schikorski and Stevens, 1997*). We found that the spine density along the dendrites of D1- and D2-MSNs was not altered in the absence of

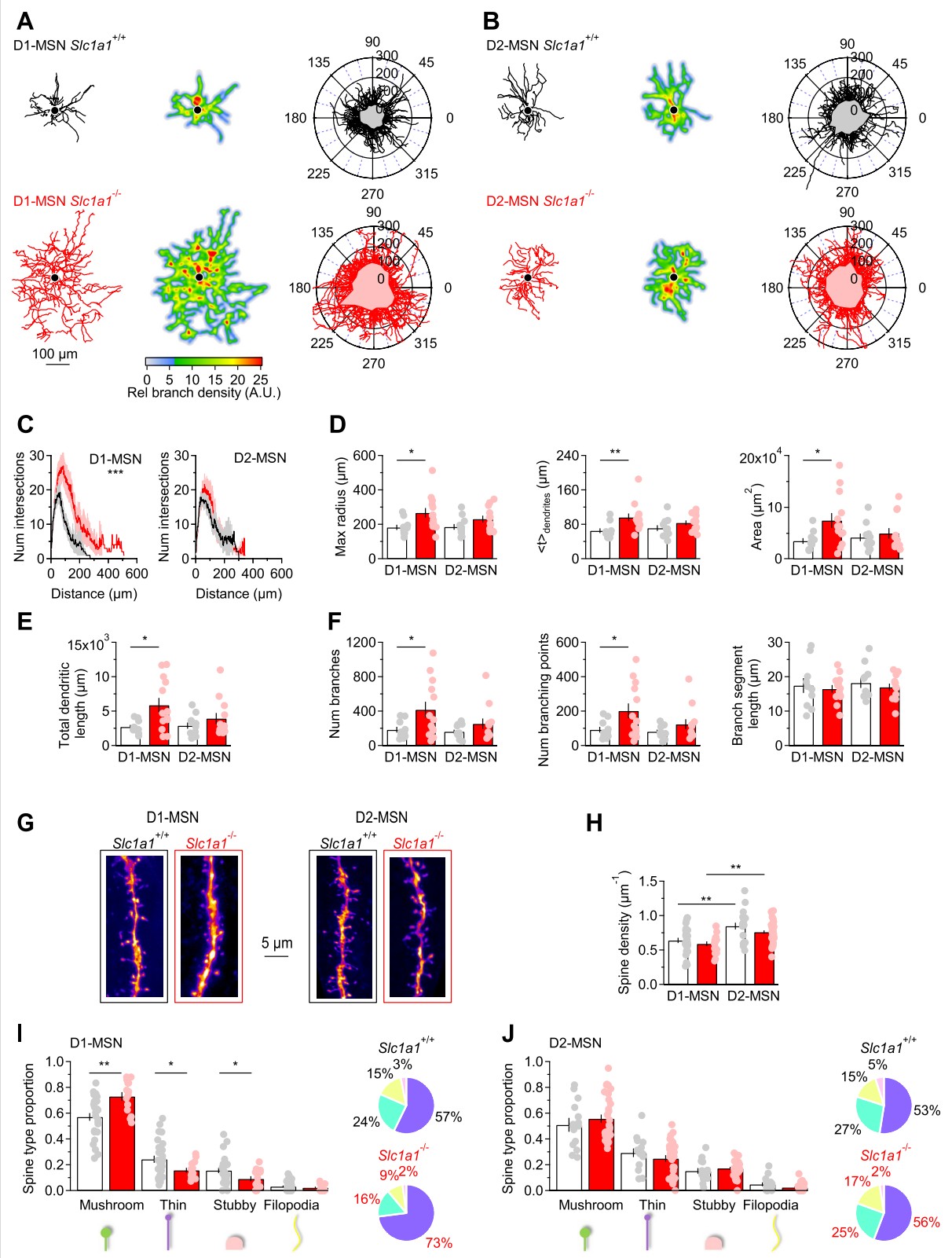

**Figure 2.** EAAC1 limits the branching pattern and spine size in D1-MSNs. (**A**) *Left,* Example of biocytin-filled and reconstructed D1-MSNs in *Slc1a1*⁺/⁺ (*top, black*) and *Slc1a1*⁻/⁻ mice (*bottom, red*). *Center,* heat maps of the reconstructions shown in the left panel. Regions highlighted in red are the areas with the highest branch density. *Right,* polar plots showing overlaid reconstructions of D1-MSNs in *Slc1a1*⁺/⁺ (n=10 neurons) and *Slc1a1*⁻/⁻ mice (n=13 neurons). The gray and pink shaded areas represent the mean coverage area of D1-MSNs in each mouse genotype. (**B**) As in **A**, for D2-MSNs in *Slc1a1*⁺/⁺

*Figure 2 continued on next page*

 Research article

Neuroscience

(n=11 neurons) and *Slc1a1⁻/⁻* mice (n=12 neurons). (**C**) *Left,* Sholl plots showing the number of dendritic branch intersections formed by D1-MSNs at increasing distances from the center of the soma, for the D1-MSNs described in A. *Right,* As in left, for D2-MSNs. (**D**) *Left,* The maximum radius of dendritic branches, calculated as the maximum x-value in the Sholl plots in C, is increased in D1-MSNs of *Slc1a1⁻/⁻* mice. *Center,* The geometric centroid (<*t* > ) is the center of mass of the Sholl plots, which is the distance from the soma before and after which there is the same density of branches (i.e., number/weight). *Right,* Analysis of DLS neuropil coverage by MSNs, showing that D1-MSNs occupy a larger domain of the DLS neuropil in *Slc1a1⁻/⁻* mice. (**E**) The total length of all dendrites is larger in D1-MSNs of *Slc1a1⁻/⁻* mice. (**F**) Dendritic branch analysis of DLS MSNs. The total number of dendritic branches (*left*) and the number of branching points (*center*) are increased in D1-MSNs of *Slc1a1⁻/⁻* compared to *Slc1a1⁺/⁺* mice. The mean branch length is similar among MSNs of *Slc1a1⁺/⁺* and *Slc1a1⁻/⁻* mice (*right*). (**G**) 2D maximum intensity projections of confocal images of dendritic branches from D1-MSNs (*left*) and D2-MSNs (*right*) in *Slc1a1⁺/⁺* (*black frame*) and *Slc1a1⁻/⁻* mice (*red frame*). (**H**) The density of dendritic spines is larger in D2-MSNs compared to D1-MSNs, in *Slc1a1⁺/⁺* and *Slc1a1⁻/⁻* mice (*Slc1a1⁺/⁺* D1-MSNs n=27 neurons, D2-MSNs n=18 neurons; *Slc1a1⁻/⁻* D1-MSN n=18 neurons, D2-MSN n=38 neurons). (**I**) Spine classification analysis, showing that there is an increase in the proportion of mushroom spines and a decrease in the proportion of thin and stubby spines in D1-MSNs from *Slc1a1⁻/⁻* mice. The pie charts on the right show the distribution of different spine types in D1-MSNs from *Slc1a1⁺/⁺* (*black*) and *Slc1a1⁻/⁻* mice (*red*). Same n values as in H. (**J**) As in I, for D2-MSNs. Different types of spines are equally represented in D2-MSNs from *Slc1a1⁺/⁺* and *Slc1a1⁻/⁻* mice. Same n values as in H. Data represent mean ± SEM.

EAAC1 (p=0.34 and p=0.12, respectively), but was higher in D2- compared to D1-MSNs in *Slc1a1⁺/⁺* and *Slc1a1⁻/⁻* mice (**p=2.4e-3 and **p=2.7e-4, respectively; *Figure 2G and H*). Given that the total dendritic length is larger in *Slc1a1⁻/⁻* mice, the presence of a similar density of spines in D1-MSNs of *Slc1a1⁺/⁺* and *Slc1a1⁻/⁻* mice indicates that D1-MSNs receive a larger number of excitatory synaptic inputs in the absence of EAAC1 (spine number = spine density · dendritic length). In addition, there was an increase in the proportion of mushroom spines and a decrease in that of thin and stubby spines in D1-MSNs of *Slc1a1⁻/⁻* mice (D1-MSN mushroom spines: **p=1.1e-3; thin spines: *p=0.01; stubby spines: *p=0.02; *Figure 2I*). This effect was not detected when comparing the spine size distribution in D2-MSNs of *Slc1a1⁺/⁺* and *Slc1a1⁻/⁻* mice (D2-MSN mushroom spines: p=0.38; thin spines: p=0.28; stubby spines: p=0.42; *Figure 2J*). Therefore, the size of the dendritic arbor of D1-MSNs is smaller when EAAC1 is expressed, and these cells receive fewer and weaker excitatory synaptic inputs.

## EAAC1 limits excitation onto D1-MSNs

An increase in the spine number and size in D1-MSNs of *Slc1a1⁻/⁻* mice might be indicative of functional changes in the quantal parameters (*N, p* and *q*), which can manifest themselves as changes in the frequency and/or kinetics of action potential-independent miniature EPSCs (mEPSCs). Therefore, we recorded mEPSCs in the presence of the voltage-gated sodium channel blocker tetrodotoxin (TTX 1 µM) and analyzed their frequency, amplitude, and kinetics. In our experiments, the mEPSC frequency was larger in D1-MSNs of *Slc1a1⁻/⁻* mice compared to *Slc1a1⁺/⁺* mice (*p=0.01; *Figure 3A and B*). This result is consistent with an increase in the number and/or release probability of excitatory synaptic contacts onto D1-MSNs in *Slc1a1⁻/⁻* mice. In *Slc1a1⁺/⁺* mice, the mEPSC frequency was larger in D2-MSNs compared to D1-MSNs (*p=0.02), whereas the frequency of these event in *Slc1a1⁻/⁻* mice was similar (p=0.51; *Figure 3A and B*). To determine whether the increased mEPSC frequency in D1-MSNs of *Slc1a1⁻/⁻* mice could be attributed exclusively to differences in *N* (the number of functional release sites) or also differences in *p* (the release probability), we measured the paired-pulse ratio (PPR) of evoked AMPA EPSCs in D1-MSNs, as this parameter is inversely proportional to the release probability, *p*. The PPR of AMPA EPSCs was similar in D1-MSNs from *Slc1a1⁻/⁻* and *Slc1a1⁺/⁺* mice (p=0.43), arguing against differences in *p* between excitatory synapses onto D1-MSNs in *Slc1a1⁻/⁻* and *Slc1a1⁺/⁺* mice. The mEPSC analysis also revealed that the mEPSC amplitude was larger in the absence of EAAC1 (**p=1.8e-3; *Figure 3C*), with no significant difference in mEPSC kinetics (rise: $F_{genotype}(1,39)=0.033$, p=0.86; $F_{cell\ type}(1,39)=0.452$, p=0.51; $F_{genotype*cell\ type}(1,39)=0.145$, p=0.71; $t_{50}$: $F_{genotype}(1,39)=0.084$, p=0.77; $F_{cell\ type}(1,39)=1.967$, p=0.17; $F_{genotype*cell\ type}(1,39)=0.004$, p=0.95; *Figure 3D*). This is consistent with an increased quantal size (*q*) at excitatory synapses onto D1-MSNs in *Slc1a1⁻/⁻* mice, and with the increased spine size in these cells. Therefore, both *N* and *q* are increased in the absence of EAAC1.

The role of EAAC1 in modulating evoked glutamatergic transmission has been previously studied in the hippocampus, where this neuronal transporter has been shown to limit spillover activation of NMDA receptors (*Diamond, 2001*; *Scimemi et al., 2009*). Whether this is a general property of EAAC1 and whether EAAC1 exerts similar effects on evoked excitatory transmission in the DLS remains unclear, given that the contribution of NMDA receptors to excitatory synaptic transmission

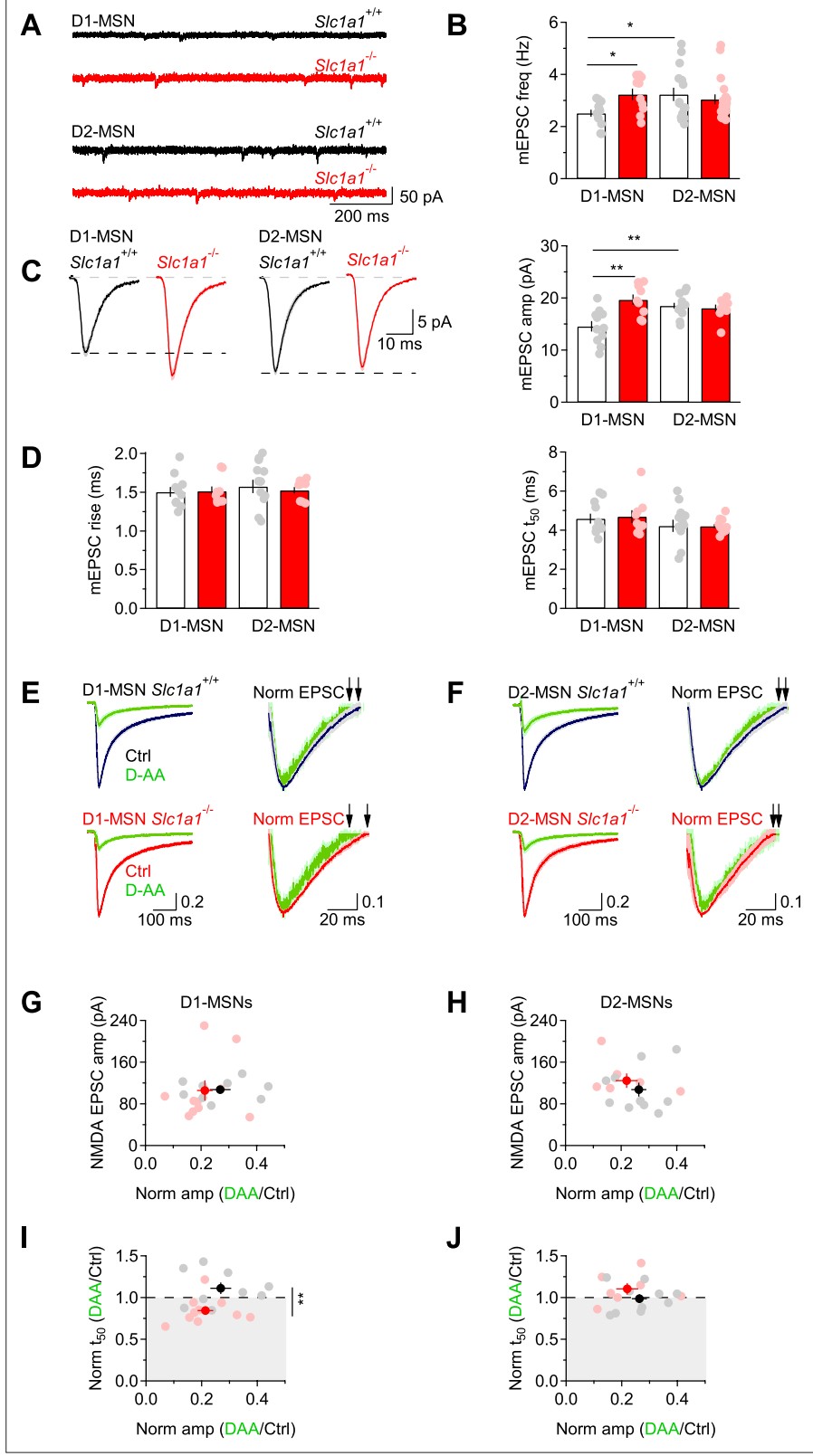

**Figure 3.** EAAC1 limits excitation in D1-MSNs. (**A**) Sample traces of mEPSCs recordings from DLS MSNs in *Slc1a1*⁺/⁺ (*black*) and *Slc1a1*⁻/⁻ mice (*red*), at V_hold=-70 mV. (**B**) Summary graph of the mEPSC frequency in DLS MSNs. In *Slc1a1*⁺/⁺ mice, the mEPSC frequency is higher in D2-MSNs (n=15 neurons) compared to D1-MSNs (n=12 neurons). In *Slc1a1*⁻/⁻ mice, there is no significant difference in the mEPSC frequency between D1-MSNs (n=10

*Figure 3 continued on next page*

*Figure 3 continued*

neurons) and D2-MSNs (n=19 neurons), but the mEPSC frequency is higher in D1-MSNs compared to *Slc1a1*$^{+/+}$ mice. Data shown in B-D were collected from the same cells. (**C**) *Left,* Average mEPSCs from *Slc1a1*$^{+/+}$ and *Slc1a1*$^{-/-}$ D1- and D2-MSNs. *Right,* Summary of the mEPSC amplitude. In *Slc1a1*$^{+/+}$ mice, the mEPSC amplitude is larger in D2- compared to D1-MSNs. In *Slc1a1*$^{-/-}$ mice, there is no significant difference in the mEPSC amplitude between D1- and D2-MSNs, but the mEPSC amplitude is larger in D1-MSNs compared to *Slc1a1*$^{+/+}$ mice. (**D**) Summary of the mEPSC rise (*left*) and 50% decay time (*right*) measured in D1- and D2-MSNs of *Slc1a1*$^{+/+}$ and *Slc1a1*$^{-/-}$ mice. (**E**) *Left,* NMDA EPSCs from D1-MSNs, recorded in Mg$^{2+}$-free extracellular solution at V$_{hold}$=-70 mV, in control conditions (*Slc1a1*$^{+/+}$ n=9 neurons: *black; Slc1a1*$^{-/-}$ n=10 neurons: *red*) and in the presence of D-AA (100 µM; *green*). *Right,* peak normalized NMDA EPSCs, shown with a y-axis range of 0.5–1. This allows visualizing the time used for the calculations of the t$_{50}$, highlighted by the black arrows. Each trace represents the average of 20 consecutive trials. (**F**) As in E, for D2-MSNs (*Slc1a1*$^{+/+}$ n=10 neurons; *Slc1a1*$^{-/-}$ n=7 neurons). (**G**) Summary scatter plot showing the amplitude of NMDA EPSCs and its reduction by D-AA in D1-MSNs of *Slc1a1*$^{+/+}$ and *Slc1a1*$^{-/-}$ mice. (**H**) As in G, for D2-MSNs. (**I**) Summary scatter plot showing that D-AA reduced the NMDA EPSC amplitude to the same extent in *Slc1a1*$^{+/+}$ and *Slc1a1*$^{-/-}$ D1-MSNs but reduced the t$_{50}$ only in D1-MSNs from *Slc1a1*$^{-/-}$ mice. (**J**) As in I, for D2-MSNs. Data represent mean ± SEM.

The online version of this article includes the following figure supplement(s) for figure 3:

**Figure supplement 1.** NMDA/AMPA ratio in striatal MSNs.

in the DLS is much less pronounced than in the hippocampus (***Sung et al., 2001***). This was confirmed by our own experiments, showing that the NMDA/AMPA ratio in D1-MSNs, scaled by the driving force of each current, was two to four times smaller than we observed in the hippocampus (***McCauley et al., 2020***; *Figure 3—figure supplement 1A, B*). If, despite the paucity of NMDA receptors in the DLS, EAAC1 limited their activation by glutamate spillover, we would expect NMDA EPSCs to decay more slowly in *Slc1a1*$^{-/-}$ compared to *Slc1a1*$^{+/+}$ mice (***Scimemi et al., 2009***). Accordingly, NMDA EPSCs recorded from D1-MSNs of *Slc1a1*$^{-/-}$ mice had similar amplitude, but longer decay in *Slc1a1*$^{-/-}$ compared to *Slc1a1*$^{+/+}$ mice (NMDA EPSC amp *Slc1a1*$^{-/-}$ vs. *Slc1a1*$^{+/+}$ p=0.70; NMDA EPSC t$_{50}$ *Slc1a1*$^{-/-}$ vs. *Slc1a1*$^{+/+}$ *p=0.02; *Figure 3—figure supplement 1C–E, right*). Although we detected a trend towards longer AMPA EPSCS decay in D1-MSNs of *Slc1a1*$^{-/-}$ compared to *Slc1a1*$^{+/+}$ mice, this did not reach statistical significance (AMPA EPSC t$_{50}$ *Slc1a1*$^{-/-}$ vs. *Slc1a1*$^{+/+}$ p=0.20; *Figure 3—figure supplement 1E, left*). This is consistent with previous works showing that AMPA receptor desensitization can mask an effect of glutamate spillover on the delayed activation of these receptors (***Scimemi and Beato, 2009***; ***Scimemi et al., 2009***; ***Hestrin et al., 1990***; ***Barbour et al., 1994***; ***Sarantis et al., 1993***; ***Diamond and Jahr, 1995***; ***Isaacson and Nicoll, 1993***).

If spillover onto D1-MSNs of *Slc1a1*$^{-/-}$ mice was increased, we would also expect NMDA EPSCs in MSNs of *Slc1a1*$^{-/-}$ mice to decay faster in the presence of the low-affinity, competitive NMDA receptor antagonist D-AA (***Diamond, 2001***). This rationale is based on evidence that: (*i*) D-AA competes with glutamate for binding to NMDA receptors (***Clements et al., 1992***), and (*ii*) D-AA preferentially blocks receptors activated by small and slow glutamate transients (i.e. peri- and extra-synaptic NMDA receptors) (***Diamond, 2001***). To test this hypothesis, we perfused brain slices with a Mg$^{2+}$-free extracellular solution, and recorded NMDA EPSCs from MSNs voltage clamped at –70 mV, a potential that allows glutamate binding and translocation via post-synaptic EAAC1 (***Wadiche et al., 1995b***; ***Arriza et al., 1994***). To make a meaningful comparison of the effect of D-AA on the NMDA EPSC decay, we used a sub-saturating concentration of D-AA and confirmed that this reduced the NMDA EPSC amplitude to the same extent in *Slc1a1*$^{+/+}$ and *Slc1a1*$^{-/-}$ mice, as this is mostly accounted for by activation of synaptic NMDA (***Clements et al., 1992***). We evoked EPSCs of similar amplitude in D1- and D2-MSNs of *Slc1a1*$^{+/+}$ and *Slc1a1*$^{-/-}$ mice (D1-MSNs *Slc1a1*$^{+/+}$ vs. *Slc1a1*$^{-/-}$: p=0.95; D2-MSNs *Slc1a1*$^{+/+}$ vs. *Slc1a1*$^{-/-}$: p=0.40; *Figure 3E–H*). D-AA (100 µM) reduced the NMDA EPSC amplitude to similar levels in D1-MSNs (*Slc1a1*$^{+/+}$: ***p=5.2e-8; *Slc1a1*$^{-/-}$: ***p=4.6e-10; *Slc1a1*$^{-/-}$ vs. *Slc1a1*$^{+/+}$ p=0.26) and D2-MSNs of *Slc1a1*$^{+/+}$ and *Slc1a1*$^{-/-}$ mice (*Slc1a1*$^{+/+}$: ***p=6.9e-10; *Slc1a1*$^{-/-}$: ***p=1.2e-6; *Slc1a1*$^{-/-}$ vs. *Slc1a1*$^{+/+}$ p=0.38; *Figure 3G–J*). Under these experimental conditions, D-AA sped the NMDA EPSC only in D1-MSNs of *Slc1a1*$^{-/-}$ mice (norm t$_{50}$ D-AA/Ctrl D1-MSNs *Slc1a1*$^{+/+}$: p=0.15; *Slc1a1*$^{-/-}$: *p=0.01; *Slc1a1*$^{-/-}$ vs. *Slc1a1*$^{+/+}$ **p=7.2e-3; norm t$_{50}$ D-AA/Ctrl D2-MSNs *Slc1a1*$^{+/+}$: p=0.82; *Slc1a1*$^{-/-}$: p=0.18; *Slc1a1*$^{-/-}$ vs. *Slc1a1*$^{+/+}$ p=0.20; *Figure 3I–J*). These findings indicate that EAAC1 limits spillover activation of NMDA receptors at excitatory synapses onto D1-MSNs, not D2-MSNs, suggesting the existence of a preferential effect of EAAC1 on these cells.

## EAAC1 strengthens inhibition onto D1-MSNs

One interesting property of EAAC1 is that its expression is not limited to glutamatergic synapses, but can also be detected in axonal boutons of GABAergic neurons, where it may serve to supply these cells with a substrate for GABA synthesis and release (*Conti et al., 1998*; *Rothstein et al., 1994*; *He et al., 2000*). This is because even though GABA can be synthesized de novo in axonal boutons, part of it can also be synthesized from glutamate recycled from the extracellular space via glutamate transporters (*Scimemi, 2014*). This recycling pathway differs in complexity depending on whether recycling of extracellular glutamate relies on neuronal or glial glutamate transporters (*Figure 4A, left*). Whereas glutamate taken up by EAAC1 is converted into GABA via decarboxylation in the presynaptic terminal, glutamate taken up by glial transporters needs first to be converted into glutamine in the astrocyte cytoplasm, which is shuttled to neurons and converted first into glutamate and ultimately into GABA (*Figure 4A, left*). There are multiple unknowns about these two recycling pathways. For example, it is not known how much GABA release onto MSNs relies on de novo synthesis versus recycling via neuronal or glial glutamate transporters. It is also unknown whether EAAC1 contributes differently to action potential-dependent and -independent GABA release. This is important, given the growing body of work indicating that vesicles mediating spontaneous and evoked inhibitory synaptic transmission are partially segregated among synapses and may utilize partially different molecular machineries (*Horvath et al., 2020*; *Sara et al., 2011*; *Wang et al., 2022*).

We addressed these concerns first by recording mIPSCs in MSNs from $Slc1a1^{+/+}$ and $Slc1a1^{-/-}$ mice. The mIPSCs were recorded in the presence of voltage-gated sodium channel, AMPA and NMDA receptor blockers (TTX 1 µM, NBQX 10 µM and APV 50 µM, respectively) from MSNs voltage clamped at 40 mV. By holding cells at this depolarized potential, we limited possible confounding effects due to post-synaptic uptake via EAAC1, which is inhibited by membrane depolarization (*Wadiche et al., 1995b*). We recorded mIPSCs before and after applying T-TBOA (1 µM), a broad-spectrum glutamate transporter antagonist (*Tsukada et al., 2005*). Our results showed that T-TBOA decreased the mIPSC amplitude in D1-MSNs from $Slc1a1^{+/+}$ mice by~15%, without changing the mIPSC kinetics ($Slc1a1^{+/+}$ D1-MSN mIPSC amplitude, Ctrl vs. T-TBOA: **p=5.9e-4; rise: p=0.33; $t_{50}$: p=0.45). By contrast, the amplitude and kinetics of mIPSCs recorded from D1-MSNs of $Slc1a1^{-/-}$ mice were not altered by T-TBOA ($Slc1a1^{-/-}$ D1-MSN mIPSC amplitude, Ctrl vs. T-TBOA: p=0.29; rise: p=0.11; $t_{50}$: p=0.55; *Figure 4B and C*). The mIPSC amplitude and kinetics in D2-MSNs were not altered by T-TBOA, in $Slc1a1^{+/+}$ and $Slc1a1^{-/-}$ mice ($Slc1a1^{+/+}$ D2-MSN mIPSC amplitude, Ctrl vs. T-TBOA: p=0.06; rise: p=0.19; $t_{50}$: p=0.20; $Slc1a1^{-/-}$ D2-MSN mIPSC amplitude, Ctrl vs. T-TBOA: p=0.25; rise: p=0.78; $t_{50}$: p=0.59; *Figure 4D and E*). Since there is evidence that spillover activation of CB1 cannabinoid and mGluRI-III receptors can also inhibit GABA release, we asked whether the reduced mIPSC amplitude in D1-MSNs detected in the presence of T-TBOA could be due, at least in part, to CB1 and/or mGluRI-III activation (*Drew et al., 2008*; *Ohno-Shosaku et al., 2002*; *Zhu and Lovinger, 2005*; *Semyanov and Kullmann, 2000*; *Mitchell and Silver, 2000*). If this were the case, we would expect T-TBOA to have a smaller effect on the mIPSC amplitude when applied in the continued presence of the CB1 inverse agonist AM251 (2 µM; *Figure 4—figure supplement 1A*), of LY341495 (100 µM) which, at this concentration, acts as a broad mGluRI-III antagonist (*Kingston et al., 1998*; *Figure 4—figure supplement 1B*), or of both drugs together (*Figure 4—figure supplement 1C*). The results of these experiments showed that T-TBOA reduced the mIPSC amplitude in D1-MSNs even in the presence of AM251 (Ctrl: ***p=1.1e-4; AM251 ***p=2.6e-5), LY341495 (**p=3.6e-3) or AM251 +LY341495 (**p=1.3e-3). Importantly, T-TBOA reduced the mIPSC amplitude to the same extent in control conditions and when CB1 and/or mGluRI-III receptors were blocked (F(1,49)=0.406, p=0.75; *Figure 4—figure supplement 1D*), ruling out a potential contribution of CB1 and mGluRI-III to the regulation of the quantal size of GABAergic mIPSCs in D1-MSNs. Together, these findings indicate that: (*i*) EAAC1 is the sole glutamate transporter contributing to filling of vesicles used for action potential-independent GABA release onto D1-MSNs; (*ii*) neither EAAC1 nor astrocytic glutamate transporters contribute to filling of vesicles released spontaneously onto D2-MSNs. These conclusions were consistent with two additional observations. *First*, in control conditions, the mIPSC amplitude, rise and decay time were smaller in D1-MSNs from $Slc1a1^{-/-}$ compared to $Slc1a1^{+/+}$ mice (mIPSC amplitude: *p=0.03; rise: *p=0.02; $t_{50}$: ***p=9.6e-5; *Figure 4B and C*). By contrast, the amplitude, rise and 50% decay time of mIPSCs recorded from D2-MSNs were similar in $Slc1a1^{+/+}$ and $Slc1a1^{-/-}$ mice (amplitude: p=0.34; rise: p=0.26; $t_{50}$: p=0.06; *Figure 4D and E*). *Second*, T-TBOA reduced the mIPSC frequency only in D1-MSNs from $Slc1a1^{+/+}$ (***p=2.0e-4), not

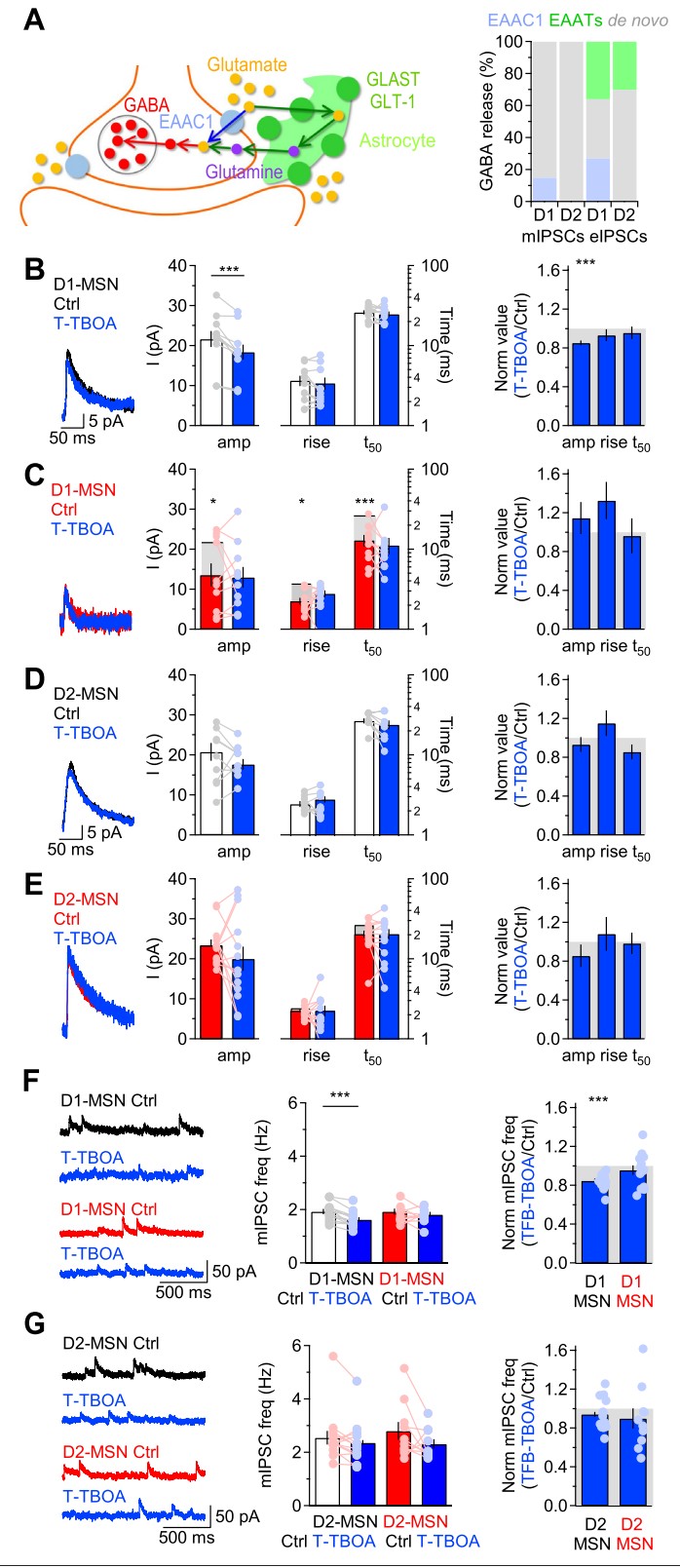

**Figure 4.** EAAC1 strengthens action potential-independent GABAergic inhibition onto D1-MSNs. (**A**)*Left,* Schematic representation for routes of glutamate uptake via neuronal and glial glutamate transporters. *Right,* Summary of the contribution of glutamate uptake via EAAC1, EAATs and de novo GABA synthesis to action potential-independent GABA release. (**B**) *Left,* Example of GABA mIPSCs recorded from D1-MSNs of *Slc1a1*⁺/⁺

*Figure 4 continued on next page*

*Figure 4 continued*

mice in control conditions (*black*) and in the presence of T-TBOA (1 µM; *blue*), at V$_{hold}$ = 40 mV. *Center,* Summary of the mIPSC amplitude and kinetics recorded before and after T-TBOA (n=11 neurons). *Right,* Summary of the relative effect of T-TBOA on the mIPSC amplitude and kinetics. T-TBOA induced a significant reduction in the amplitude of mIPSCs recorded from D1-MSNs. (**C–E**) As in B, for DLS D1-MSNs of *Slc1a1$^{-/-}$* mice (n=10 neurons), and D2-MSNs of *Slc1a1$^{+/+}$* mice (n=15 neurons), and *Slc1a1$^{-/-}$* mice (n=13 neurons), respectively. In panels C, E we also included data collected from *Slc1a1$^{+/+}$* mice to show that loss of EAAC1 leads to smaller and faster mIPSCs in D1-MSNs, whereas no significant difference is detected between D2-MSNs of *Slc1a1$^{+/+}$* and *Slc1a1$^{-/-}$* mice. (**F**) Summary of the effect of T-TBOA on the mIPSC frequency in D1-MSNs of *Slc1a1$^{+/+}$* and *Slc1a1$^{-/-}$* mice, with representative traces (*left*), raw data (*center*) and normalized values (*right*). (**G**) As in F, for D2-MSNs. Data represent mean ± SEM.

The online version of this article includes the following figure supplement(s) for figure 4:

**Figure supplement 1.** T-TBOA reduces action potential-independent GABAergic inhibition onto D1-MSNs in the presence of CB1 and mGluRI-III receptor antagonists.

---

*Slc1a1$^{-/-}$* mice (p=0.22; *Figure 4F*). There was no significant effect of T-TBOA on the mIPSC frequency in D2-MSNs (*Slc1a1$^{+/+}$*: p=0.13; *Slc1a1$^{-/-}$*: p=0.14; *Figure 4G*). Therefore, EAAC1 shapes not only glutamatergic transmission but also action potential-independent GABA release onto D1-MSNs.

We next analyzed the effect of T-TBOA on evoked IPSCs (*Figure 5A–D*). T-TBOA reduced the IPSC amplitude in MSNs of *Slc1a1$^{+/+}$* and *Slc1a1$^{-/-}$* mice, without altering their rise and 50% decay time (*Slc1a1$^{+/+}$* D1-MSN Ctrl vs. T-TBOA amplitude: \*\*p=1.6e-3, rise: p=0.08, t$_{50}$: p=0.79; *Slc1a1$^{-/-}$* D1-MSN Ctrl vs. T-TBOA amplitude: \*p=0.03, rise: p=0.88, t$_{50}$: p=0.07; *Slc1a1$^{+/+}$* D2-MSN Ctrl vs. T-TBOA amplitude: \*p=0.03, rise: p=0.11, t$_{50}$: p=0.15; *Slc1a1$^{-/-}$* D2-MSN Ctrl vs. T-TBOA amplitude: \*\*p=7.2e-3, rise: p=0.21, t$_{50}$: p=0.10). Overall, T-TBOA reduced the IPSC amplitude in D1-MSNs of *Slc1a1$^{+/+}$* mice more than in any other type of MSNs tested in the experiments (*Figure 5E–H*). The reduction of the IPSC amplitude induced by T-TBOA was ~63% in D1-MSNs of *Slc1a1$^{+/+}$* mice and~36% in D1-MSNs of *Slc1a1$^{-/-}$* mice (\*p=0.02; *Figure 5A, B and E*). We confirmed that these results were not biased by a potential presynaptic effect of T-TBOA on release probability at GABAergic synapses, because T-TBOA did not change the IPSC PPR (PPR Ctrl vs. T-TBOA *Slc1a1$^{+/+}$* D1-MSN: p=0.19; *Slc1a1$^{-/-}$* D1-MSN: p=0.85; *Slc1a1$^{+/+}$* D2-MSN: p=0.40; *Slc1a1$^{-/-}$* D2-MSN: p=0.16; *Figure 5A–D and H*). We confirmed that T-TBOA reduced the IPSC amplitude in D1-MSNs even when applied in the presence of AM251 (\*\*\*p=7.4e-4; *Figure 5—figure supplement 1A*), LY341495 (\*\*p=1.3e-3; *Figure 5—figure supplement 1B*) or AM251+LY341495 (\*\*\*p=5.4e-4; *Figure 5—figure supplement 1C*). These effects were not significantly different from those obtained when T-TBOA was applied in control conditions (F(1,29)=0.800, p=0.50; *Figure 5—figure supplement 1D*) and were not associated with changes in the IPSC PPR (PPR *Slc1a1$^{+/+}$* D1-MSN AM251 vs. T-TBOA: p=0.52; LY341495 vs. T-TBOA: p=0.54; AM251+LY341495 vs. T-TBOA: p=0.14; *Figure 5—figure supplement 1A–C, right*).

Together, these results suggest that EAAC1 contributes to ~27% of the quantal size of vesicles released in an action potential-dependent manner onto D1-MSNs (i.e. 63%–36%), with a~36% contribution by other glial transporters and ~37% contribution by de novo synthesis (i.e. 100%–63%; *Figure 4A, right*). In D2-MSNs, T-TBOA reduced the IPSC amplitude by ~30% in *Slc1a1$^{+/+}$* and *Slc1a1$^{-/-}$* mice, suggesting that GABA supply for evoked release onto D2-MSNs relies for ~70% on de novo synthesis and for ~30% on glial glutamate transporters, with no contribution of EAAC1 (*Figure 4A, right*; *Figure 5C–E*). Therefore, EAAC1 enhances spontaneous and evoked GABAergic transmission onto D1-MSNs, without altering GABAergic inhibition onto D2-MSNs.

## EAAC1 strengthens reciprocal inhibition between D1-MSNs

MSNs form extensive collaterals within the DLS. Accordingly, the dendritic field of each MSN extends over a volume that contains >2800 MSNs, and forms ~1200–1800 synaptic contacts onto these cells (*Wilson, 2007*). Functional studies suggest that each D1-MSN has a connection rate of 26% and 6% with other D1- and D2-MSNs, respectively (*Taverna et al., 2008*; *Tecuapetla et al., 2009*). By contrast, each D2-MSN has a connection rate of 28% and 36% with other D1- and D2-MSNs, respectively (*Tecuapetla et al., 2009*; *Taverna et al., 2008*). The presence of such an intricate inhibitory network of connections is thought to be important for regulating the excitability of MSNs and, more generally, the output of the striatum and the execution of coordinated motor behaviors (*Burke et al., 2017*;

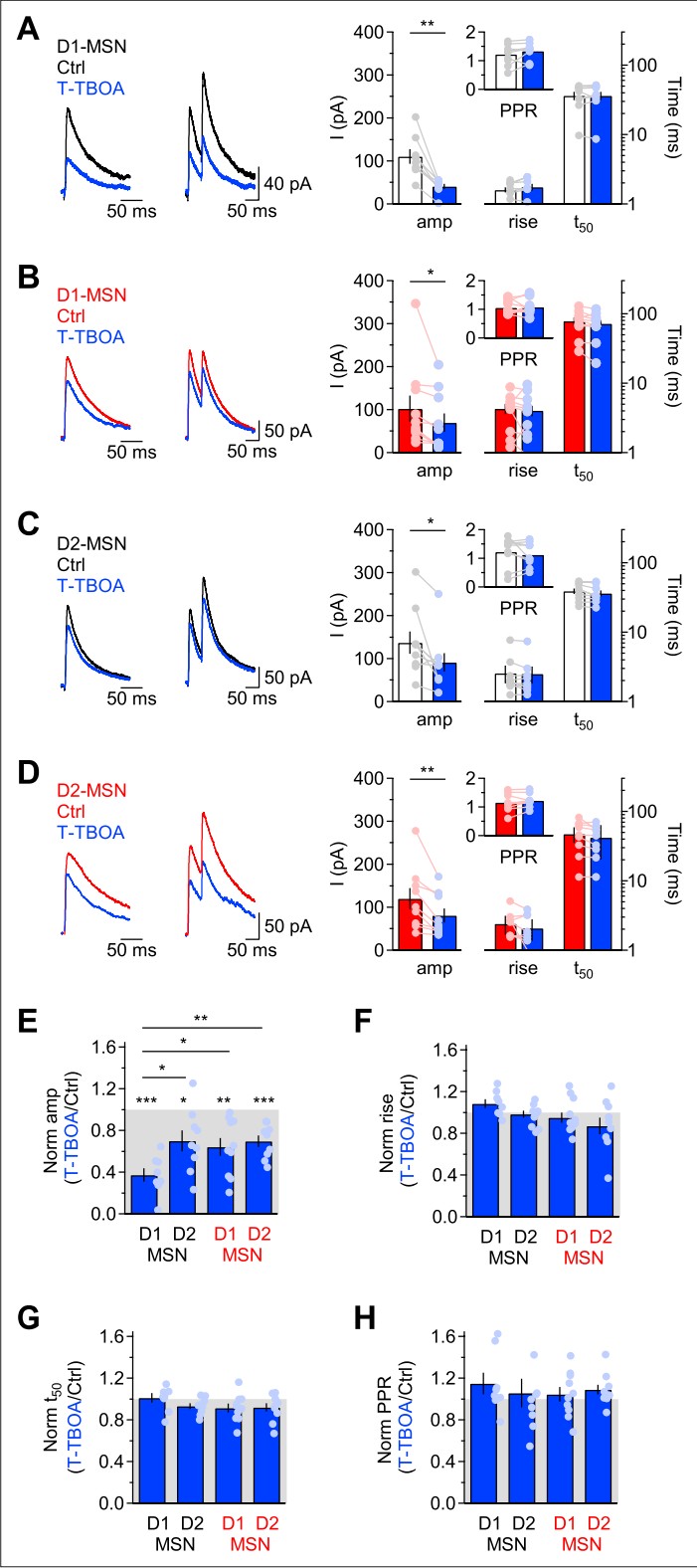

**Figure 5.** EAAC1 strengthens action potential-dependent GABAergic inhibition onto D1-MSNs. (**A**)*Left and center,* Example of evoked single and paired IPSCs recorded from D1-MSNs of *Slc1a1*$^{+/+}$ mice in control conditions (*black*) and in the presence of T-TBOA (1 μM; *blue*), at V$_{hold}$ = 40 mV. *Right,* Summary of the IPSC amplitude and kinetics recorded before and after T-TBOA (n=8 neurons). The inset represents the summary values of the PPR. (**B–D**) As in A, for DLS D1-MSNs of *Slc1a1*$^{-/-}$ mice (n=10 neurons) and D2-MSNs of *Slc1a1*$^{+/+}$ mice (n=9 neurons), and *Slc1a1*$^{-/-}$

*Figure 5 continued on next page*

*Figure 5 continued*

mice (n=9 neurons), respectively. (**E–H**) Summary of the relative effect of T-TBOA on the IPSC amplitude and kinetics. T-TBOA induced a significant reduction on the IPSC amplitude, but not of the rise time, 50% decay time and PPR. T-TBOA reduced the IPSC amplitude in D1-MSNs of $Slc1a1^{+/+}$ mice more than in any other type of MSNs tested in the experiments. Data represent mean ± SEM.

The online version of this article includes the following figure supplement(s) for figure 5:

**Figure supplement 1.** T-TBOA reduces action potential-dependent GABAergic inhibition onto D1-MSNs in the presence of CB1 and mGluRI-III receptor antagonists.

---

*Dobbs et al., 2016*). To determine to which extent EAAC1 shapes lateral inhibition among different types of MSNs, we performed bilateral stereotaxic injections of a conditional ChR2-expressing AAV in the DLS of $Drd1a^{Cre/+}$:$Rosa26^{tm9/tm9}$ or $Adora2a^{Cre/+}$:$Rosa26^{tm9/tm9}$ mice (**Figure 6A, left**). Three weeks later, we prepared acute coronal slices, patched D1- or D2-MSNs, identified for their expression of the tdTomato reporter, and used blue light stimuli for full-field activation of ChR2 expressed in either D1- or D2-MSNs (**Figure 6A, right**). We set the intensity of the blue light stimulation to a power of ~250 μW, measured at the sample plane (**Figure 6—figure supplement 1A**). Each stimulation lasted 5ms, a duration that did not evoke any adaptation in post-synaptic, ChR2-mediated photo-currents (**Figure 6—figure supplement 1B**), and was repeated every 30 s, to allow for full recovery of ChR2 from activation (**Figure 6—figure supplement 1D**). The ChR2 photocurrents, isolated in the presence of TTX (1 μM), reversed at ~20 mV (**Figure 6—figure supplement 1C**) and their amplitude did not change in the presence of the GABA$_A$ receptor antagonist picrotoxin (100 μM). Therefore, we used 20 mV as the holding potential to record optogenetically-evoked IPSCs (oIPSCs), isolated pharmacologically by adding NBQX (10 μM) and APV (50 μM) to the external solution. Under these experimental conditions, the oIPSCs were completely blocked by the GABA$_A$ receptor antagonist picrotoxin (100 μM). To confirm the accuracy of the stereotaxic injections in the DLS, at the completion of the recordings, we fixed the slices and imaged them using a confocal microscope (**Figure 6B**). Our experiments showed that the oIPSC amplitude was smaller at D1-D1 synapses from $Slc1a1^{-/-}$ compared to $Slc1a1^{+/+}$ mice, suggesting that EAAC1 might enhance synaptic inhibition at these synapses (*p=0.04; **Figure 6D and H**). By contrast, the amplitude of oIPSCs evoked at D2-D1 synapses was similar across the two genotypes (p=0.65; **Figure 6E, I**). The oIPSC amplitude was also similar at D1-D2 (p=0.79; **Figure 6F and J**) and D2-D2 synapses of $Slc1a1^{+/+}$ and $Slc1a1^{-/-}$ mice (p=0.54; **Figure 6G and K**). Although T-TBOA (1 μM) reduced the oIPSC amplitude at all collateral MSN synapses, its effect was largest at D1-D1 synapses (**Figure 6D–K**). Accordingly, the reduction of the oIPSC amplitude induced by T-TBOA at D1-D1 synapses was ~53% in $Slc1a1^{+/+}$ mice and ~29% in $Slc1a1^{-/-}$ mice (**p=9.1e-3; **Figure 6H**). This suggests that GABAergic inhibition between D1-MSNs relies for ~24% on glutamate uptake via EAAC1 (i.e. 53%–29%), and for ~29% on glutamate uptake mediated by glial transporters (**Figure 6C**). The reduction of the oIPSC amplitude induced by T-TBOA was similar at D2-D1 (p=0.72), D1-D2 (p=0.26) and D2-D2 synapses in $Slc1a1^{+/+}$ and $Slc1a1^{-/-}$ mice (p=0.86; **Figure 6D–K**). These findings suggest that glial glutamate transporters contribute to 33–44% of GABA release across different types of collateral synapses formed between MSNs (**Figure 6C**).

Although the FISH data suggest that EAAC1 is mostly expressed in MSNs (**Figure 1**), we asked whether it could also shape GABAergic inhibition onto D1-MSNs originating from other cell types. In addition to receiving reciprocal inhibition from D1/2-MSNs, D1-MSNs receive strong feedforward inhibition from a class of interneurons, the fast-spiking parvalbumin interneurons (PV-INs), which In vivo can fire action potentials at high frequency (*Burke et al., 2017*; *Tepper et al., 2004*; *Tepper et al., 2010*; *Tepper et al., 2008*). To determine whether EAAC1 also altered GABAergic inhibition from PV-INs onto D1-MSNs, we performed whole-cell patch clamp recordings from tdTomato-expressing D1-MSNs in mice that received stereotaxic injections of a viral construct that allowed expression of the red-shifted opsin C1V1 in PV-INs (PHP.eB-S5E2-C1V1-eYFP; **Figure 6L and M**; *Vormstein-Schneider et al., 2020*). In these experiments, T-TBOA reduced the oIPSC amplitude similarly in $Slc1a1^{+/+}$ and $Slc1a1^{-/-}$ mice (**p=5.7e-3 and *p=0.04, respectively; $Slc1a1^{+/+}$ vs. $Slc1a1^{-/-}$: p=0.15; **Figure 6N and O, bottom**). This finding suggests that EAAC1 does not contribute significantly to GABA release at PV to D1-MSN synapses (**Figure 6C**).

Together, these results indicate that glial glutamate transporters enhance inhibition at homo- and hetero-synaptic contacts formed between different types of MSNs and, to a lesser extent, at synapses

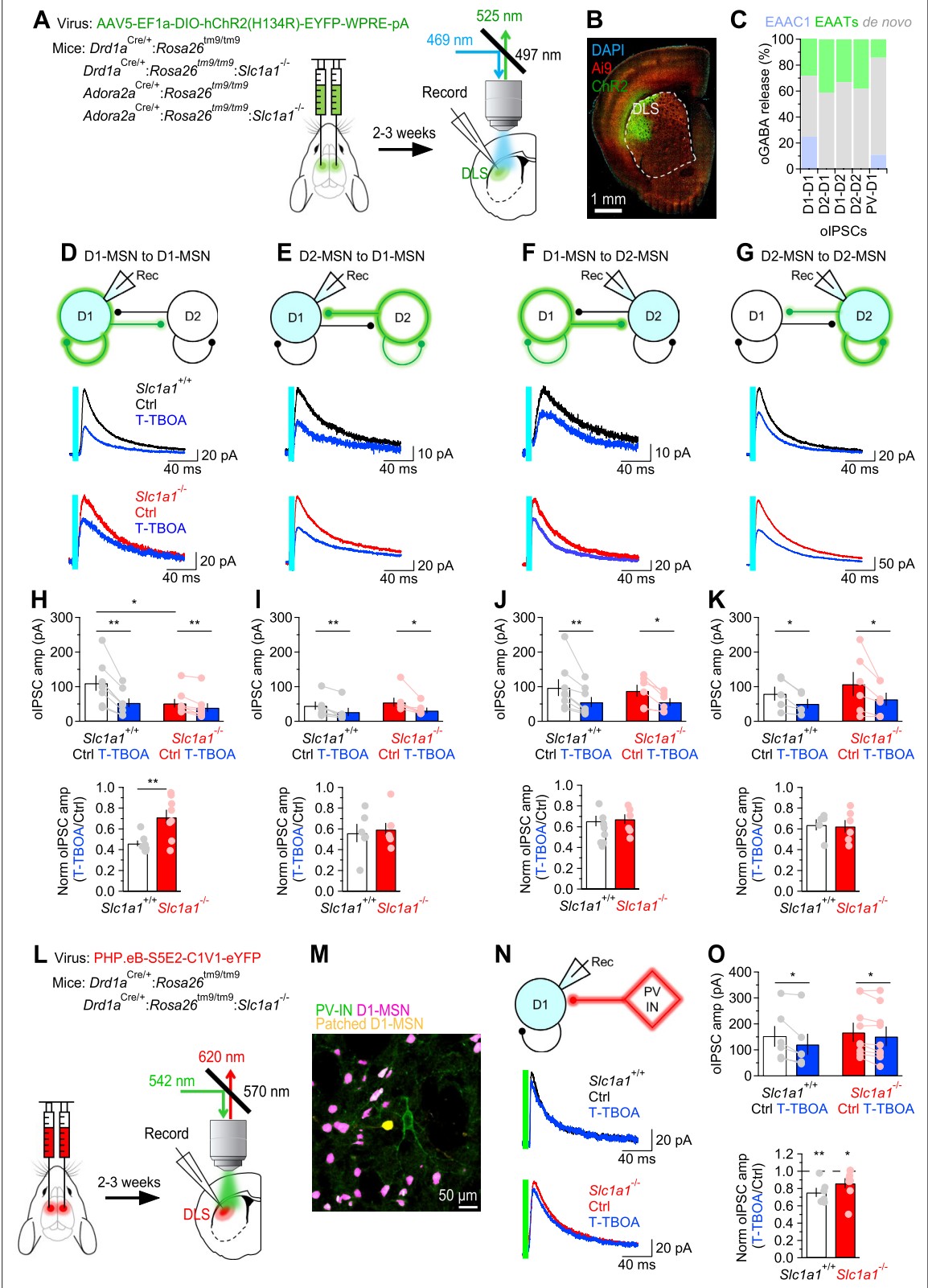

**Figure 6.** EAAC1 strengthens GABAergic inhibition at D1-D1 synapses. (**A**) Schematics of viral transduction in the DLS and light evoked stimulation of ChR2-transfected MSNs in slices. (**B**) Confocal image of mouse coronal slice transfected with ChR2. (**C**) *Right,* Summary of the contribution of glutamate uptake via EAAC1, EAATs and de novo GABA synthesis to action potential-dependent GABA release evoked by light activation of ChR2 at different sets of synapses. (**C**) Summary of the contribution of different types of glutamate transporters and de novo GABA synthesis to GABA released during oIPSCs.

*Figure 6 continued on next page*

Figure 6 continued

(**D**) *Top,* Schematic representation of the experimental design. The patched cell was voltage clamped at $V_{hold}$ = 20 mV, the reversal potential of ChR2 photocurrents. *Middle,* Example of oIPSCs recorded from D1-MSNs in response to optogenetic stimulation of GABA release from other D1-MSNs, in mice expressing EAAC1. Each trace represents the average of 20 consecutive trials. *Bottom,* As in the middle panel, for $Slc1a1^{-/-}$ mice. (**E–G**) As in D, for D2-D1 (**E**), D1-D2 (**F**) and D2-D2 synapses (**G**). (**H**) *Top,* In-cell comparison of D1-D1 oIPSCs in $Slc1a1^{+/+}$ (n=8 neurons) and $Slc1a1^{-/-}$ mice (n=8 neurons), before and after T-TBOA (1 μM; *blue*). *Bottom,* Summary of the peak normalized oIPSC amplitude. (**I**) As in H, for D2-D1 oIPSCs in $Slc1a1^{+/+}$ (n=6 neurons) and $Slc1a1^{-/-}$ mice (n=7 neurons). (**J**) As in H, for D1-D2 oIPSCs in $Slc1a1^{+/+}$ (n=8 neurons) and $Slc1a1^{-/-}$ mice (n=6 neurons). (**K**) As in H, for D2-D2 oIPSCs in $Slc1a1^{+/+}$ (n=5 neurons) and $Slc1a1^{-/-}$ mice (n=6 neurons). (**L**) Schematics of viral transduction in the DLS and light evoked stimulation of C1V1-transfected PV-INs in slices. (**M**) Confocal image of tdTomato expressing D1-MSNs (*magenta*) and C1V1 transfected PV-INs (*green*) in a slice from which we recorded oIPSCs from a D1-MSN (*yellow*). (**N**) As in D, for oIPSCs recorded from D1-MSNs in response to green light activation of C1V1 expressed in PV-INs. (**O**) As in H, for oIPSCs recorded in response to C1V1 activation of GABA release from PV-INs to D1-MSNs ($Slc1a1^{+/+}$ n=6 neurons; $Slc1a1^{-/-}$ n=9 neurons). Data represent mean ± SEM.

The online version of this article includes the following figure supplement(s) for figure 6:

**Figure supplement 1.** Optimization of stimulation parameters for optogenetic stimulation.

formed between PV-INs and D1-MSNs (*Figure 6C*). By contrast, EAAC1 significantly enhances synaptic inhibition only at homosynaptic GABAergic contacts formed between D1-MSNs.

## EAAC1 limits the firing output of D1-MSNs

An essential aspect of information processing is the ability to transform synaptic inputs into action potential outputs, allowing the DLS to control the activity of its target regions. The data obtained so far provide an opportunity to shed light on the functional role of homosynaptic, reciprocal inhibition between D1-MSNs. For this reason, we asked how the changes in synaptic excitation and reciprocal inhibition between D1-MSNs induced by EAAC1 shape the firing output of these cells. To answer this, we used the NEURON platform and performed modeling experiments on reconstructed D1-MSNs from $Slc1a1^{+/+}$ and $Slc1a1^{-/-}$ mice, to account for the different morphological features of these cells (*Figure 2*; *Hines and Carnevale, 1997*). In these simulations, the passive and active membrane properties of D1-MSNs were consistent with those of D1-MSNs in $Slc1a1^{+/+}$ and $Slc1a1^{-/-}$ mice (*Figure 7—figure supplement 1*), and the weight of each excitatory and inhibitory input was set to be consistent with our experimental data (*Figure 7—figure supplements 2–4*). We allowed each D1-MSN to receive 100 excitatory and 100 inhibitory inputs, randomly distributed along the dendrites, activated at a range of frequencies consistent with the range of synaptic inputs received by the striatum in vivo. We analyzed the effect of these parameter manipulations on firing rates of D1-MSNs across the theta (~8 Hz) and beta range (~20 Hz), also representative of the firing activity detected in the striatum in vivo (*Figure 7A and B*; *Berke et al., 2004*; *Pennartz et al., 2009*). We then asked whether and how EAAC1 altered the offset and the gain in the input-output relationship of D1-MSNs (*Figure 7C*, *left and center*). Briefly, changing the offset allows D1-MSNs to subtract basal levels of synaptic activity, altering the range of input stimulation frequencies that evoke firing (an additive/subtractive operation). Changing the gain alters the sensitivity of D1-MSNs to varying levels of synaptic input rates, while preserving the range over which these inputs evoke spiking activity (a multiplicative/divisive operation; *Mitchell and Silver, 2003*). In the absence of synaptic inhibition, EAAC1 induced a modest increase in offset, which became more pronounced as the inhibition rate increased (*Figure 7D*). With increasing rates of inhibition onto D1-MSNs, EAAC1 also decreased the gain, and therefore switched from having a purely additive effect to also having a divisive effect (*Figure 7D*, *right*). Conversely, when excitation was low, EAAC1 reduced the basal firing rate of D1-MSNs, with a negligible effect on gain (*Figure 7E*, *left*). As excitation increased, the increase in gain becomes more evident and is associated with a slight decrease in the basal firing rate of D1-MSNs (*Figure 7E*, *right*). The overall effect of EAAC1 across a broader range of activity of E/I is to introduce a frequency-dependent increase in offset and a frequency-independent decrease in gain (*Figure 7C*, *right*). Together, these results suggest that EAAC1 can perform both subtractive and divisive operations, depending on the rate of incoming E/I onto D1-MSNs. The higher is the rate of synaptic E/I onto D1-MSNs, the greater is the reduction of the firing output of D1-MSNs by EAAC1. By modulating E/I, EAAC1 narrows the dynamic range and reduces the sensitivity of D1-MSNs to incoming inputs.

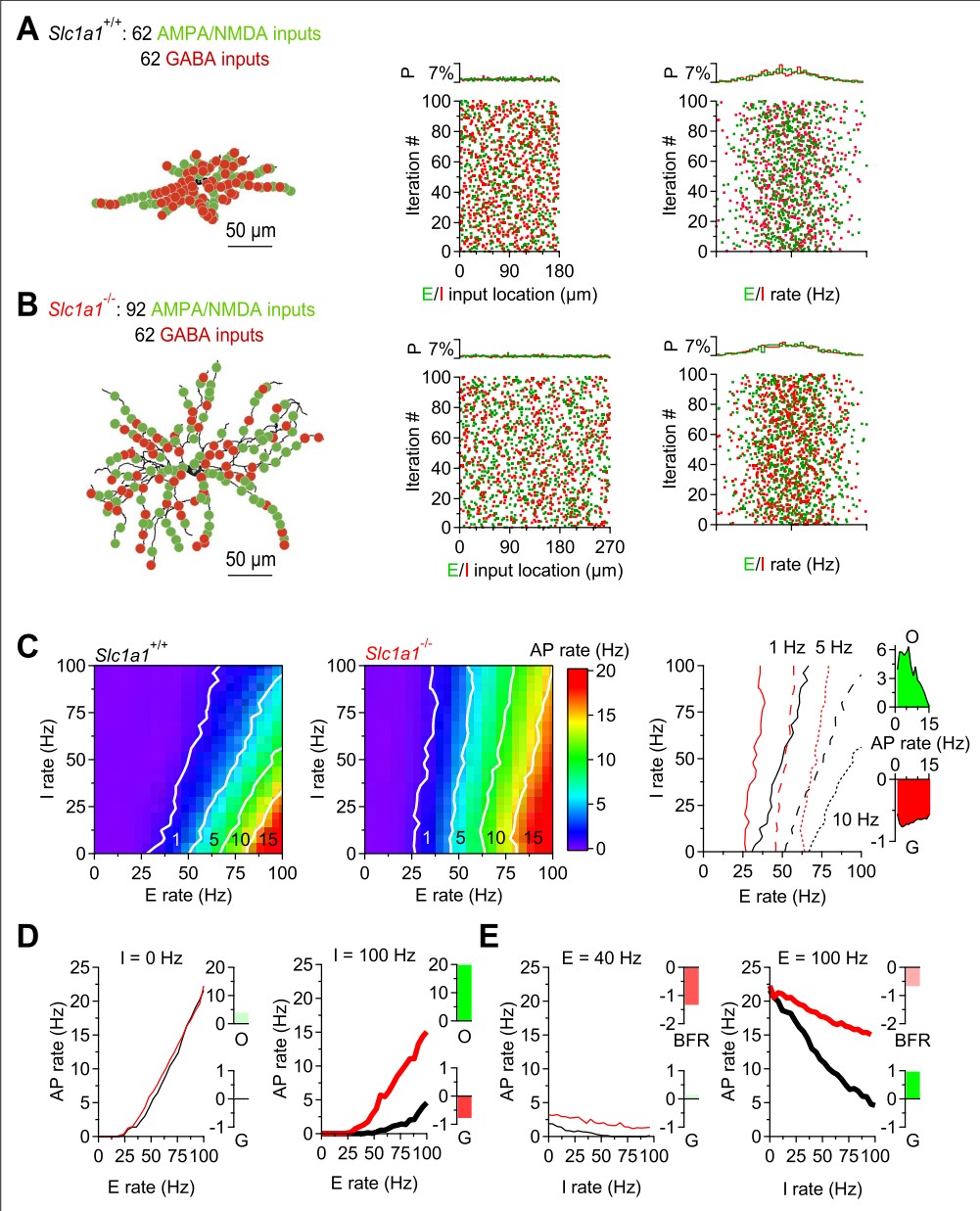

**Figure 7.** EAAC1 reduces the firing output of D1-MSNs in a realistic compartmental model of D1-MSNs. (**A**)*Left,* Representation of a biocytin-filled D1-MSN in *Slc1a1*[+/+] mice, with excitatory (*green*) and inhibitory inputs (*red*) randomly distributed along all dendrites. *Center,* Spatial distribution of excitatory and inhibitory inputs along the length of the dendrites, *Right,* Schematic representation of the instantaneous frequency of activation of excitatory and inhibitory inputs in each of the 100 simulation iterations. The activation frequency for these inputs was set to have a Gaussian distribution. (**B**) As in A, for a biocytin-filled D1-MSN in *Slc1a1*[-/-] mice. (**C**) *Left,* Heat map showing the firing output of D1-MSNs in *Slc1a1*[+/+] mice for different combinations of activity for excitatory (*x-axis*) and inhibitory inputs (*y-axis*). *Center*, as in the right panel, for *Slc1a1*[-/-] mice. *Right,* Overlaid contour lines for action potential firing at 1, 5, and 10 Hz in D1-MSNs from *Slc1a1*[+/+] and *Slc1a1*[-/-] mice. (**D**) *Left,* Output firing frequency-input frequency relationships for synaptic excitation for D1-MSNs of *Slc1a1*[+/+] (*black*) and *Slc1a1*[-/-] mice (*red*), in the absence (*thin curves*) of synaptic inhibition. *Right,* As in the left panel for high inhibition levels (i.e. 100 Hz; *thick curves*). (**E**) As in D, to compare the effect of inhibition at varying levels of excitation in D1-MSNs of *Slc1a1*[+/+] and *Slc1a1*[-/-] mice. Data represent means from 100 simulations.

The online version of this article includes the following figure supplement(s) for figure 7:

**Figure supplement 1.** Optimization of ball-and-stick NEURON model of D1-MSN excitability.

*Figure 7 continued on next page*

*Figure 7 continued*

**Figure supplement 2.** Optimization of synaptic weight for excitatory synapses containing AMPA receptors, using a NEURON model of D1-MSNs with realistic morphology.

**Figure supplement 3.** Optimization of synaptic weight for excitatory synapses containing NMDA receptors, using a NEURON model of D1-MSNs with realistic morphology.

**Figure supplement 4.** Optimization of synaptic weight for inhibitory synapses containing GABA$_A$ receptors, using a NEURON model of D1-MSNs with realistic morphology.

## The role of excitation and reciprocal inhibition between D1-MSNs in reward-based behaviors

The activity of MSNs is important for the execution of coordinated movements and reward-based behaviors (*Freeze et al., 2013*; *Kravitz et al., 2010*; *Lobo et al., 2010*; *Kravitz et al., 2012*). Although heterosynaptic inhibition from D2-MSNs has been shown to inhibit action potential firing in D1-MSNs (*Dobbs et al., 2016*), much less is known about the functional significance of homosynaptic inhibition between D1-MSNs. Computational models suggest that this form of inhibition might be involved in driving coherence or synchronizing functional units of information processing in the striatum, known as ensembles (*Humphries and Prescott, 2010a*; *Humphries et al., 2010b*; *Humphries et al., 2009*; *Moyer et al., 2014*; *Ponzi and Wickens, 2012*, *Ponzi and Wickens, 2010*; *Ponzi and Wickens, 2013*; *Yim et al., 2011*). If the striatum truly operates as a collection of ensembles driving specific behaviors, one may hypothesize that the reduction in synaptic excitation and reciprocal inhibition among D1-MSNs in *Slc1a1*$^{-/-}$ mice might increase their propensity to switch between different reward-based behaviors. We tested this hypothesis using a simple probabilistic reward lever press task, in which we trained mice to receive a water reward at different probability ($P_{rew}$ = 0.5‖0.9; *Figure 8A–E*). In the training sessions, in both *Slc1a1*$^{+/+}$ and *Slc1a1*$^{-/-}$ mice, the number of collected rewards was proportional to the reward probability, and inversely related to the number of lever presses (rewards *Slc1a1*$^{+/+}$ $P_{rew}$=0.5 vs. $P_{rew}$ = 0.9: (F1,30)=14.880, \*\*p=1.6e-3; rewards *Slc1a1*$^{-/-}$ $P_{rew}$ = 0.5 vs. $P_{rew}$ = 0.9: (F1,26)=14.403, \*p=2.2e-3; lever presses *Slc1a1*$^{+/+}$ $P_{rew}$=0.5 vs. $P_{rew}$ = 0.9: (F1,30)=36.524, \*\*\*p=2.2e-5; lever presses *Slc1a1*$^{-/-}$ $P_{rew}$ = 0.5 vs. $P_{rew}$ = 0.9: (F1,26)=29.173, \*\*\*p=1.2e-4; *Figure 8B and D*). No significant difference between *Slc1a1*$^{+/+}$ and *Slc1a1*$^{-/-}$ mice was detected when $P_{rew}$ = 0.5‖0.9 (rewards *Slc1a1*$^{+/+}$ vs *Slc1a1*$^{-/-}$ at $P_{rew}$ = 0.5: (F1,28)=2.910, p=0.10; rewards *Slc1a1*$^{+/+}$ vs. *Slc1a1*$^{-/-}$ at $P_{rew}$ = 0.9: (F1,28)=1.301, p=0.26; lever presses *Slc1a1*$^{+/+}$ vs *Slc1a1*$^{-/-}$ at $P_{rew}$ = 0.5: (F1,28)=3.187, p=0.09; lever presses *Slc1a1*$^{+/+}$ vs *Slc1a1*$^{-/-}$ at $P_{rew}$ = 0.9: (F1,28)=0.691, p=0.41 *Figure 8C and E*). We then ran test sessions where the reward probability was changed every 5–75 s (*Figure 8F–N*). In these experiments, we still detected an inverse relationship between the number of rewards/lever presses and the reward probability, suggesting that mice can detect quick changes in reward probability and can rapidly switch between different reward-based behaviors over the whole range of tested switch times (i.e. 5–75 s; rewards *Slc1a1*$^{+/+}$ at $P_{rew}$ = 0.5 vs. $P_{rew}$ = 0.9: F(1,30)=17.19, \*\*\*p=2.6e-4; rewards *Slc1a1*$^{-/-}$ at $P_{rew}$ = 0.5 vs. $P_{rew}$ = 0.9: F(1,26)=5.095, \*p=0.03; lever presses *Slc1a1*$^{+/+}$ at $P_{rew}$ = 0.5 vs. $P_{rew}$ = 0.9: F(1,30)=14.023, \*\*\*P=7.7e-4; lever presses *Slc1a1*$^{-/-}$ at $P_{rew}$ = 0.5 vs. $P_{rew}$ = 0.9: F(1,26)=4.260, \*p=0.04; *Figure 8K and M*). In these experiments, *Slc1a1*$^{+/+}$ and *Slc1a1*$^{-/-}$ mice collected a similar number of rewards and performed a similar number of lever presses when the switch time was <15 s (rewards *Slc1a1*$^{-/-}$ vs. *Slc1a1*$^{+/+}$ at $P_{rew}$ = 0.5, $t_{switch}$ <15 s: F(1,28)=0.061, p=0.81; rewards *Slc1a1*$^{-/-}$ vs. *Slc1a1*$^{+/+}$ at $P_{rew}$ = 0.9, $t_{switch}$ <15 s: F(1,28)=0.082, p=0.78; lever presses *Slc1a1*$^{+/+}$ vs. *Slc1a1*$^{-/-}$ at $P_{rew}$ = 0.5, $t_{switch}$ <15 s: F(1,28)=0.048, p=0.83; lever presses *Slc1a1*$^{+/+}$ vs. *Slc1a1*$^{-/-}$ at $P_{rew}$ = 0.9, $t_{switch}$ <15 s: F(1,28)=0.040, p=0.84; *Figure 8L and N*). However, as the switch time increased above 30 s, *Slc1a1*$^{-/-}$ mice outperformed *Slc1a1*$^{+/+}$ mice, collecting more rewards at low and high reward probabilities (rewards *Slc1a1*$^{-/-}$ vs. *Slc1a1*$^{+/+}$ at $P_{rew}$ = 0.5, $t_{switch}$ >30 s: F(1,28)=8.687, \*\*p=6.4e-3; rewards *Slc1a1*$^{-/-}$ vs. *Slc1a1*$^{+/+}$ at $P_{rew}$ = 0.9, $t_{switch}$ >30 s: F(1,28)=9.514, \*\*p=4.6e-3; lever presses *Slc1a1*$^{+/+}$ vs. *Slc1a1*$^{-/-}$ at $P_{rew}$ = 0.5, $t_{switch}$ >30 s: F(1,28)=9.511, \*\*p=4.6e-3; lever presses *Slc1a1*$^{+/+}$ vs. *Slc1a1*$^{-/-}$ at $P_{rew}$ = 0.9, $t_{switch}$ >30 s: F(1,28)=9.270, \*\*p=5.0e-3; *Figure 8L and N*). This suggests that neuronal circuits modulated by EAAC1, capable of altering excitation and reciprocal inhibition onto D1-MSNs, only limit the execution of slowly switching behaviors.

If these results are due to loss of EAAC1 from D1-MSNs, we would expect the behavior of *Drd1a*$^{Cre/+}$:*Slc1a1*$^{f/f}$ mice (which do not express EAAC1 in D1-MSNs) to be comparable to that of

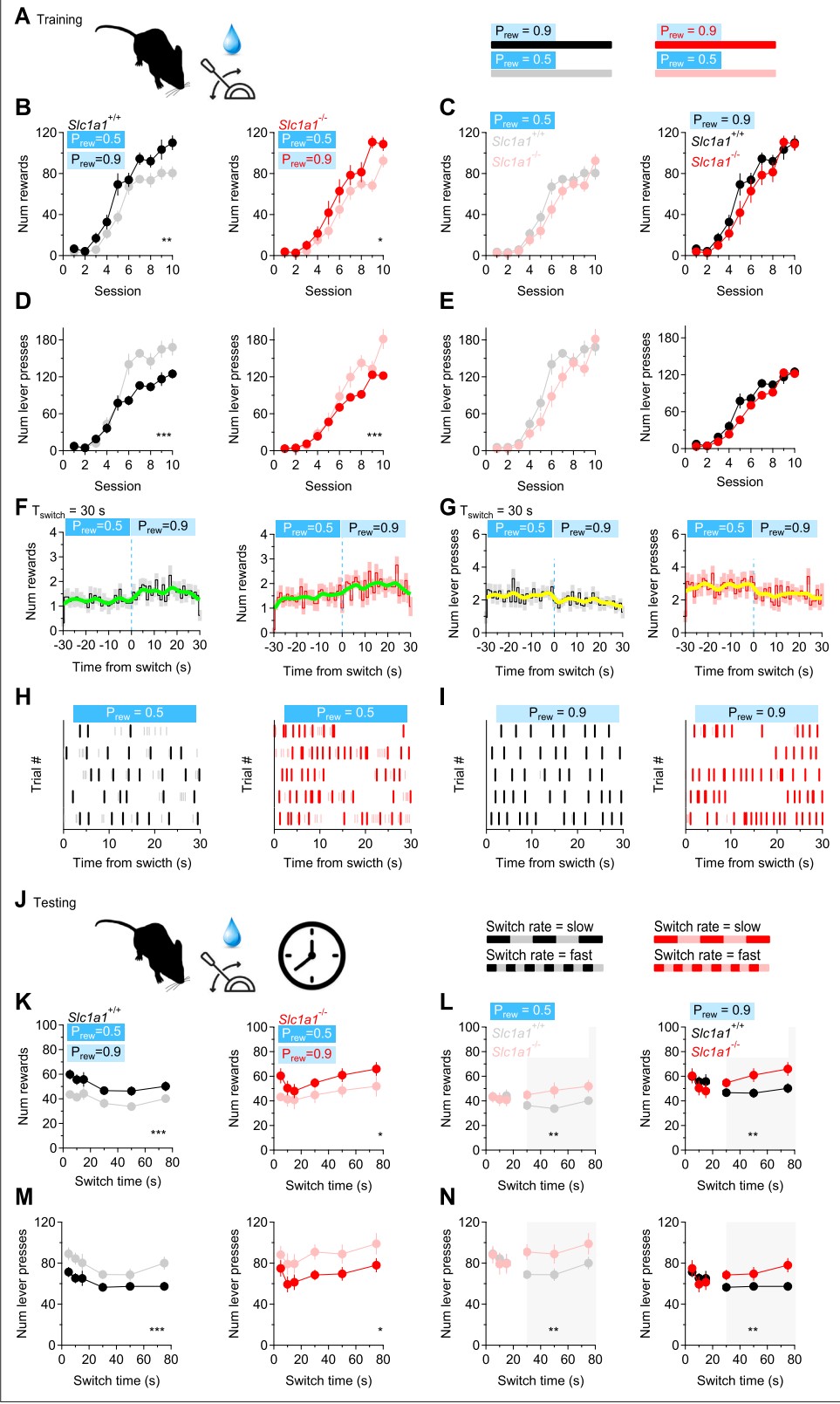

**Figure 8.** EAAC1 shapes slow-switching behaviors. (**A**) Schematic representation of the training sessions, with water rewards delivered at two different reward probabilities (P_rew), during each 5 min session. (**B**) Number of rewards collected by *Slc1a1*[+/+] (n=16 mice) and *Slc1a1*[-/-] mice (n=14 mice), at P_rew = 0.5 and P_rew = 0.9 over the course of 10 training sessions. (**C**) Comparison between *Slc1a1*[+/+] and *Slc1a1*[-/-] mice for the data shown in B. (**D,**

*Figure 8 continued*

E) As in B, C, showing the total number of lever presses performed by the mice. (**F**) Temporal distribution of the number of rewards collected by *Slc1a1*$^{+/+}$ (*left*) and *Slc1a1*$^{-/-}$ mice (*right*) when the reward probability was switched from P$_{rew}$ = 0.5 to P$_{rew}$ = 0.9. The thick green line represents a binomial smoothing of the mean. (**G**) As in F, for the number of lever presses. (**H**) Raster plot showing the temporal distribution of lever presses in a 5 min long trial, in which the reward probability was switched from P$_{rew}$ = 0.5 to P$_{rew}$ = 0.9 every 30 s. The raster plots in this panel were collected at P$_{rew}$ = 0.5. Color-coded tick marks (*gray/pink*) represent the times when the lever was pressed. Other tick marks (*black/red*) represent the times when the water rewards were collected. (**I**) As in H, for P$_{rew}$ = 0.9. (**J**) Schematic representation of the test session, with water rewards delivered at two different values of P$_{rew}$, switching at different time intervals (5–75 s). (**K–N**) As in B-E, for the test session. Data represent mean ± SEM.

The online version of this article includes the following figure supplement(s) for figure 8:

**Figure supplement 1.** EAAC1 expression in D1-MSNs shapes slow-switching behaviors.

**Figure supplement 2.** EAAC1 expression in D1-MSNs of the DLS and VMS shapes slow-switching behaviors.

**Figure supplement 3.** EAAC1 expression in the SN/VTA does not alter slow-switching behaviors.

---

*Slc1a1*$^{-/-}$ mice (*Figure 8—figure supplement 1*). By contrast, we would expect the behavior of *Adora2a*$^{Cre/+}$:*Slc1a1*$^{f/f}$ mice (which do not express EAAC1 in D2-MSNs) to be similar to that of *Slc1a1*$^{+/+}$ mice (*Figure 8—figure supplement 1*). Consistent with these hypotheses, the results of experiments performed on *Drd1a*$^{Cre/+}$:*Slc1a1*$^{f/f}$ and *Adora2a*$^{Cre/+}$:*Slc1a1*$^{f/f}$ mice showed that when the switch time was short (<15 s), *Drd1a*$^{Cre/+}$:*Slc1a1*$^{f/f}$ and *Adora2a*$^{Cre/+}$:*Slc1a1*$^{f/f}$ mice collected a similar number of rewards (rewards *Drd1a*$^{Cre/+}$:*Slc1a1*$^{f/f}$ vs. *Adora2a*$^{Cre/+}$:*Slc1a1*$^{f/f}$ at P$_{rew}$ = 0.5, t$_{switch}$ <15 s: F(1,30)=2.138, p=0.15; rewards *Drd1a*$^{Cre/+}$:*Slc1a1*$^{f/f}$ vs. *Adora2a*$^{Cre/+}$:*Slc1a1*$^{f/f}$ at P$_{rew}$ = 0.9, t$_{switch}$ <15 s: F(1,30)=1.032, p=0.32; *Figure 8—figure supplement 1L*) and performed a similar number of lever presses (lever presses *Drd1a*$^{Cre/+}$:*Slc1a1*$^{f/f}$ vs. *Adora2a*$^{Cre/+}$:*Slc1a1*$^{f/f}$ at P$_{rew}$ = 0.5, t$_{switch}$ <15 s: F(1,30)=2.801, p=0.11; lever presses *Drd1a*$^{Cre/+}$:*Slc1a1*$^{f/f}$ vs. *Adora2a*$^{Cre/+}$:*Slc1a1*$^{f/f}$ at P$_{rew}$ = 0.9, t$_{switch}$ <15 s: F(1,30)=0.892, p=0.35; *Figure 8—figure supplement 1N*). As the switch time increased (30–75 s), *Drd1a*$^{Cre/+}$:*Slc1a1*$^{f/f}$ mice collected more rewards than *Adora2a*$^{Cre/+}$:*Slc1a1*$^{f/f}$ mice, at low and high reward probabilities (rewards *Drd1a*$^{Cre/+}$:*Slc1a1*$^{f/f}$ vs. *Adora2a*$^{Cre/+}$:*Slc1a1*$^{f/f}$ at P$_{rew}$ = 0.5, t$_{switch}$ >30 s: F(1,30)=9.741, \*\*p=4.0e-3; rewards *Drd1a*$^{Cre/+}$:*Slc1a1*$^{f/f}$ vs. *Adora2a*$^{Cre/+}$:*Slc1a1*$^{f/f}$ at P$_{rew}$ = 0.9, t$_{switch}$ >30 s: F(1,30)=8.031, \*\*p=8.1e-3; *Figure 8—figure supplement 1L, N*). Overall, the phenotype of *Drd1a*$^{Cre/+}$:*Slc1a1*$^{f/f}$ mice in these experiments was similar to that of *Slc1a1*$^{-/-}$ mice, whereas that of *Adora2a*$^{Cre/+}$:*Slc1a1*$^{f/f}$ mice was similar to that of *Slc1a1*$^{+/+}$ mice (*Figure 8* and *Figure 8—figure supplement 1*). This suggests that loss of expression of EAAC1 from D1-MSNs is sufficient to recapitulate the task switching behavior of *Slc1a1*$^{-/-}$ mice.

Although the DLS is recruited during task switching, reward-based flexibility in executive control also relies on neuronal activity in ventral medial regions of the striatum (VMS; *Wallis, 2007*; *Gu et al., 2008*). Given the presence of an abundant population of D1-MSNs in the VMS, we asked whether removing EAAC1 from D1-MSNs in the VMS could also allow mice to engage more effectively in slowly switching reward-based behaviors. To test this hypothesis, we repeated the task switching test in *Slc1a1*$^{f/f}$ mice that received stereotaxic injections of a Cre-dependent viral construct (AAV-D1Cre) either in the DLS or VMS, respectively (*Figure 8—figure supplement 2*). Mice lacking EAAC1 in DLS or VMS D1-MSNs displayed task switching behaviors similar to those of *Slc1a1*$^{-/-}$ mice (*Figure 8—figure supplement 2*) suggesting that EAAC1 expression in D1-MSNs throughout the striatum limits the execution of reward-based behavior flexibility for slowly switching time intervals.

Lastly, we asked whether loss of EAAC1 in D1-MSNs outside the striatum could also recapitulate the behavior of *Slc1a1*$^{-/-}$ mice. To address this question, we repeated the injections of AAV-D1Cre in the *substantia nigra* (*pars compacta* and *reticulata*; SN) and ventral tegmental area (VTA) of *Slc1a1*$^{f/f}$ mice, where D1 receptors are also abundantly expressed (*Cadet et al., 2010*; *Savasta et al., 1986*; *Boyson et al., 1986*; *Wamsley et al., 1989*; *Figure 8—figure supplement 3*). The behavior of these mice in the operant task switching test was similar to that of *Slc1a1*$^{+/+}$ mice, pointing to a specific role of EAAC1 in striatal D1-MSNs in the execution of slowly switching reward-based behaviors.

Together, these findings suggest that increased excitation onto D1-MSNs and reciprocal inhibition between striatal, but not nigral/tegmental D1-MSNs, limits execution of reward-based behaviors and compulsivity during slow action switching.

## Discussion

The main finding in this work is that the neuronal glutamate transporter EAAC1 exerts a cell-preferential control of E/I in the DLS: it limits excitation onto D1-MSNs and enhances homosynaptic lateral inhibition between D1-MSNs. By doing so, EAAC1 increases the offset and decreases the gain of the firing output of D1-MSNs. These effects are associated with reduced ability to engage in slowly switching reward-based behaviors.

### EAAC1 regulates D1-MSN morphology

To the best of our knowledge, the data presented here provide the first indication that the neuronal glutamate transporter EAAC1 is implicated with the control of neuron and spine morphology. Our previous analysis in the hippocampus indicated that the spine density and spine head diameter are not different in CA1 pyramidal cells of *Slc1a1*$^{+/+}$ and *Slc1a1*$^{-/-}$ mice (*Scimemi et al., 2009*). However, this work did not include a Sholl analysis or a detailed analysis of spine type distribution, like the one included in this manuscript. It is not known whether the effects we detect here are induced specifically by EAAC1 or whether they could also be induced by a comparable loss of glial glutamate transporters. Although there is evidence that the expression of glial glutamate transporters is altered in neuropsychiatric disorders, and that this is associated with a reduction in spine density and dendritic arborization (*O'Donovan et al., 2017*), a causal link between these structural effects and disease onset remains to be established. It might be tempting to speculate that the increase in spine size and dendritic complexity in D1-MSNs of *Slc1a1*$^{-/-}$ mice might be due to increased spillover onto NMDA receptors, since there is evidence that glutamate triggers morphological changes in neurons and promotes the acquisition of a mature spine morphology largely by acting on these receptors (*Mattison et al., 2014*; *Kwon and Sabatini, 2011*). If this were true, one could hypothesize that the lack of effect of genetic loss of EAAC1 on D2-MSNs might be due to the inability of EAAC1 to alter synaptic transmission onto these cells. This, however, would not be supported by the results obtained in the hippocampus, where EAAC1 limits spillover onto NMDA receptors, yet has no detectable effect on spine size (*Scimemi et al., 2009*). It is also possible that the effects on the morphology of D1-MSNs may be a downstream effect of EAAC1 on other cells, although interestingly dopamine depletion induces spine pruning in D2- more than D1-MSNs (*Witzig et al., 2020*; *Gagnon et al., 2017*; *Day et al., 2006*; *Suárez et al., 2014*; *Toy et al., 2014*; *Fieblinger et al., 2014*). While these findings indicate that the signaling pathways implicated in the control of spine and dendrite size are many and complex, they also point out to the fact that EAAC1 might be a critical, unsuspected component of these regulatory mechanisms.

### The role of EAAC1 in controlling synaptic integration in D1-MSNs

The DLS, one of the four main subdivisions of the striatum, integrates synaptic inputs from multiple brain regions to control the execution of habitual behaviors (*Hunnicutt et al., 2016*; *Wall et al., 2013*). In mice, ~1/3 of excitatory inputs to the DLS originates from the thalamus (*Huerta-Ocampo et al., 2014*), and the remaining 2/3 from the frontal associative and sensorimotor cortex (*Hunnicutt et al., 2016*; *Guo et al., 2015*; *Wall et al., 2013*; *Pan et al., 2010*). All these inputs provide excitation onto both D1- and D2-MSNs, but sensory inputs preferentially innervate D1-MSNs, whereas motor inputs preferentially target D2-MSNs (*Wall et al., 2013*). Our findings do not distinguish whether EAAC1 limits excitation at some or all these glutamatergic projection types to the DLS, but multiple lines of evidence indicate that only excitation onto D1-MSNs is affected by EAAC1. *First,* there is a reduced spillover-activation of NMDA receptors in these cells (*Figure 3*). *Second,* EAAC1 expression is associated with a reduced spine size, quantal size (*q*) and quantal content (*N*) in D1-MSNs (*Figures 2 and 3*). *Third,* the dendritic arbor of D1-MSNs is smaller when EAAC1 is expressed (*Figure 2*). Since D1-MSNs preferentially process sensorimotor information, these findings suggest that, in the DLS, EAAC1 is primarily involved with sensorimotor input integration.

How excitation onto D1-MSNs is processed, relayed to other brain regions and ultimately converted into different behavioral outputs depends on multiple factors, including: (*i*) the spatial distribution and activation time of these thalamo-cortical excitatory inputs; (*ii*) the local connectivity and pattern of activity of inhibitory connections formed by striatal interneurons and other MSNs; (*iii*) the intrinsic excitability properties of these cells, which however do not change in the absence of EAAC1 (*Figure 7—figure supplement 1*; *Burke et al., 2017*; *Preston et al., 1980*; *Somogyi*

*et al., 1981*; *Wilson and Groves, 1980*; *Kawaguchi et al., 1990*; *Yung et al., 1996*; *Park et al., 1980*). MSNs receive strong peri-somatic feedforward inhibition from fast spiking PV-INs, which is not altered by EAAC1 (*Burke et al., 2017*). A large population of 1200–1800 inhibitory inputs, however, targets the distal dendrites of these cells, and is formed by axon collaterals of other MSNs. Although these distal inputs are thought to be at a positional disadvantage and weaker than peri-somatic inhibitory inputs, they can change local synaptic integration and the firing output of MSNs (*Tunstall et al., 2002*; *Czubayko and Plenz, 2002*; *Dobbs et al., 2016*; *Koós and Tepper, 1999*; *Wilson, 2007*). Accordingly, simultaneous activation of D2-MSNs suppresses D1-MSN firing (*Dobbs et al., 2016*). Given that the coordinated activity of MSNs accounts for the complementary roles of these cells in the temporal control of movement execution (*Kravitz et al., 2010*; *Cui et al., 2013*), an increased spiking of D1-MSNs caused by altered E/I in the absence of EAAC1 could provide an important mechanism to disrupt the coordinated recruitment of MSNs during senso-rimotor integration and stereotyped movement execution. This hypothesis is supported by our own previous work, showing that *Slc1a1*[-/-] mice display an increased grooming frequency (*Bellini et al., 2018*).

## Physiological implications of offset- and gain-modulation by EAAC1

Understanding the input-output transformations of D1-MSNs is a key step in tying together the effect of EAAC1 on E/I. In the input-output relationships, changes in offset or gain exert different effects on information processing (*Abbott et al., 1997*; *Salinas and Thier, 2000*; *Schwartz and Simoncelli, 2001*; *Prescott and De Koninck, 2003*; *Mehaffey et al., 2005*). Changes in offset alter the input detection threshold (*Pavlov et al., 2009*). Changes in gain alter the sensitivity of a neuron to changes in excitatory inputs, and the range of inputs that can be discriminated (*Pavlov et al., 2009*). Excit-atory afferents to the striatum generate action potentials at different rates. In the absence of inhibitory inputs, EAAC1 increases the offset of D1-MSNs (i.e. it increases their detection threshold), but preserves the gain of their input-output relationship. As the rate of D1-D1 inhibition increases, EAAC1 increases the offset even further, while also reducing the gain (i.e. the sensitivity to changes in exci-tation rates). That is to say that the input detection threshold of D1-MSNs for thalamo-cortical excit-atory inputs is increased and the gain is slightly reduced at increasing levels of reciprocal inhibition among these cells, when EAAC1 is expressed. This flexible offset- and gain-control mechanism, which varies with the frequency of E/I and limits D1-MSNs firing in response to ongoing thalamo-cortical excitation, is lost in *Slc1a1*[-/-] mice.

Multiple studies suggest that MSNs in the DLS are organized into ensembles that can control the execution of different behaviors through their coordinated activation (*Adler et al., 2012*; *Carrillo-Reid et al., 2011*; *Carrillo-Reid et al., 2008*; *Barbera et al., 2016*). One of the conceptual models for the mechanisms of action of these functional units posits that D1-MSNs in units driving a specific behavior and D2-MSNs in units inhibiting competing behaviors are simultaneously active (*Mink, 1996*). Reciprocal inhibition between D1-MSNs would then limit the execution of competing actions, whereas reciprocal inhibition between D2-MSNs suppresses the inhibition of the desired behavior (*Mink, 1996*). According to this working model, disrupting reciprocal inhibition between D1-MSNs, as it happens in *Slc1a1*[-/-] mice, would promote unsynchronized activity across different functional units (*Humphries and Prescott, 2010a*; *Moyer et al., 2014*; *Ponzi and Wickens, 2012*, *Ponzi and Wickens, 2010*; *Ponzi and Wickens, 2013*; *Yim et al., 2011*). Would this cause the DLS to remain in a given state and promote the sustained execution of a given behavior, or would it perhaps allow easier switcing between different behaviors? Our experiments support a model in which reciprocal inhibition among D1-MSNs limits action switching only at long time intervals, and likely promotes coordinated movement execution (*Figure 8*). Consistent with this hypothesis, loss-of-function mutations of EAAC1 have been implicated with a neuropsychiatric disease characterized by loss of movement coordi-nation, compulsions and impulsivity, like OCD (*Porton et al., 2013*). Therefore, by modulating E/I onto D1-MSNs, EAAC1 may be a key regulator of the activity of functional units in the striatum also implicated with this disease. These mechanisms are particularly important in the context of neuropsy-chiatric diseases like OCD, as they might contribute to increased impulsivity and hypersensitivity to triggers, respectively (*Grassi et al., 2015*; *Boisseau et al., 2012*; *Bari and Robbins, 2013*; *Benatti et al., 2014*; *Ettelt et al., 2007*).

## Different circuits support the execution of fast and slow switching tasks

An interesting finding highlighted by our experiments is that the perception of time is important for recruiting different neuronal circuits to support cognitive flexibility. In this context, the fact that the task switching behavior of *Slc1a1*^-/- mice differs from that of *Slc1a1*^+/+ mice only at long time intervals (>30 s) suggests that excitation and lateral inhibition between D1-MSNs are engaged or change their properties slowly compared to other circuits that are preferentially involved in the execution of rapid switching tasks. Multiple open questions remain. For example, we do not know: (*i*) why is D1-D1 inhibition is recruited slowly? (*ii*) which other circuits, within or outside the striatum, support rapid task switching? Although our study does not provide a clear answer to these questions, evidence from previous works shows that: (*i*) the dopamine tone in the striatum ramps up slowly and smoothly, and alters the speed of temporal processing during timing tasks (*Rao et al., 2001*; *Matell et al., 2003*; *Howe et al., 2013*; *Westbrook and Braver, 2016*); (*ii*) striatal neurons encode temporal information slowly, over the course of many seconds, via time-dependent ramping of action potential firing (*Emmons et al., 2017*; *Bakhurin et al., 2017*; *Matell et al., 2003*; *Mello et al., 2015*; *Wang et al., 2018*; *Bruce et al., 2021*). Therefore, one could speculate that slowly evolving dopamine ramps might contribute to the recruitment of lateral D1-D1 inhibition in both *Slc1a1*^+/+ and *Slc1a1*^-/- mice, but they may exert different effects on slow task switching behaviors (i.e. different number of lever presses/rewards) due to differences in the offset- and gain-control properties of D1-MSNs due to altered excitation and lateral inhibition between D1-MSNs in *Slc1a1*^+/+ and *Slc1a1*^-/- mice. The identity and location of the circuits responsible for controlling fast task switching programs (e.g. striatal or cortical ones), at present, remains unknown.

## Functional implications of EAAC1 in different brain regions and in the context of compulsive behaviors

Increasing evidence implicates EAAC1 in the modulation of synaptic transmission in at least two regions of the brain: the hippocampus and DLS (*Bellini et al., 2018*; *Scimemi et al., 2009*; *Diamond, 2001*; *Mathews and Diamond, 2003*). In our studies, we noticed commonalities in the mechanisms through which EAAC1 controls synaptic transmission in the hippocampus and DLS, because in both regions EAAC1 acts to limit glutamate escape at excitatory synapses and increases GABA release. However, the functional consequences of these effects can vary, due to existing differences in the molecular landscape of excitatory and inhibitory synapses across brain regions. In CA1 pyramidal cells of the mouse hippocampus, EAAC1 limits glutamate escape onto NMDA receptors and promotes long-term plasticity (*Diamond, 2001*; *Scimemi et al., 2009*). In the DLS, EAAC1 also limits glutamate escape onto NMDA receptors. In addition, by preventing activation of mGluRI receptors, it promotes D1 dopamine receptor expression and long-term plasticity (*Bellini et al., 2018*). EAAC1 contributes to GABA synthesis and release from *stratum oriens* interneurons to CA1 pyramidal cells (*Mathews and Diamond, 2003*), and only at a subset of inhibitory synapses mediating lateral inhibition across D1-MSNs in the striatum. Overall, our data indicate that despite being expressed in D1- and D2-MSNs, EAAC1 exerts a preferential effect on different types of synaptic inputs targeting D1-MSNs (*Bellini et al., 2018*). This is confirmed also by our behavioral assays, showing that loss of EAAC1 in D2-MSNs does not change the ability of mice to engage in switching reward-based tasks (*Figure 8—figure supplement 1*). By contrast, loss of EAAC1 in D1-MSNs recapitulates the behavior of *Slc1a1*^-/- mice (*Figure 8—figure supplement 1*). Although our investigation of changes in cell morphology and synaptic function was limited to the DLS, we cannot rule out the possibility that analogous changes in neuron morphology and synaptic transmission may also take place in the VMS, given the similar behavioral phenotype of mice lacking EAAC1 in D1-MSNs of the DLS and VMS (*Figure 8—figure supplement 2*). Since the behavioral phenotype of mice lacking EAAC1 in D1-MSNs of the SN/VTA is similar to that of *Slc1a1*^+/+ mice (*Figure 8—figure supplement 3*), our findings point out to a cell-type specific and location-specific role of EAAC1 expression in striatal D1-MSNs with respect to compulsive-like behaviors, likely mediated via the regulation of lateral inhibition between these cells.

There are indications that there are structural changes in the hippocampus of patients affected by compulsions (including OCD, ADHD, and ASD patients; *Reess et al., 2018*; *Al-Amin et al., 2018*). Although their implications remain unclear, these findings support emerging evidence that limbic structures like the hippocampus might be implicated with compulsive behaviors in a variety

of neuropsychiatric disorders (*Milad and Rauch, 2012*; *Menzies et al., 2008a*; *Wood and Ahmari, 2015*; *Ullrich et al., 2018*).

Obviously, it is important to remember that although animal studies provide strong evidence for dysfunction in glutamatergic transmission at striatal thalamo-cortical synapses in human patients with compulsions, several other genes in the glutamatergic system, in addition to *SLC1A1*, are associated with different neuropsychiatric diseases (*Welch et al., 2007*; *Shmelkov et al., 2010*). For example, other candidate genes for OCD belong to the serotonergic and dopaminergic systems. Furthermore, non-genetic, environmental risk factors are also crucial for the manifestation of symptoms for OCD and other neuropsychiatric diseases associated with compulsions, making them complex polygenic and multifactorial diseases (*Pauls et al., 2014*; *Hoffman and Cano-Ramírez, 2022*; *Chen et al., 2022*). Despite this apparently daunting scenario, there is considerable agreement on the implication of cortico-striatal hyperactivity in compulsivity, suggesting the existence of a possible convergence of different genetic and epigenetic factors in the control of striatal function. Therefore, the findings described here may provide an example of circuit and synaptic dysfunctions shared across different genetic models of neuropsychiatric disease.

# Materials and methods

## Key resources table

| Reagent type (species) or resource | Designation | Source or reference | Identifiers | Additional information |
|---|---|---|---|---|
| Strain, strain background (*Mus musculus*) | *Slc1a1$^{-/-}$* | PMID:19923291 | | |
| Strain, strain background (*Mus musculus*) | *Slc1a$^{tm1c/tm1c}$* | This paper | | Here referred to as *Slc1a1$^{f/f}$* |
| Strain, strain background (*Mus musculus*) | *Gt(ROSA)26S or $^{tm9(CAG-tdTomato)Hze}$* | JAX | IMSR_JAX:007909 | Here referred to as *Rosa26$^{tm9/tm9}$* |
| Strain, strain background (*Mus musculus*) | *Drd1a$^{tdT/+}$* | JAX | IMSR_JAX:016204 | |
| Strain, strain background (*Mus musculus*) | *Drd1a$^{Cre/+}$* | MMRRC | MMRRC_030778-UCD | |
| Strain, strain background (*Mus musculus*) | *Adora2a$^{Cre/+}$* | MMRRC | MMRRC_036158-UCD | |
| Transfected construct (*M. musculus*) | AAV2/5-EF1a-DIO-hChR2(H134R)-EYFP-WPRE-pA | University of North Carolina | | Refers to viral construct used for stereotaxic injections |
| Transfected construct (*M. musculus*) | AAV2/9-D1-Cre-EGFP-WPRE-hGH-pA | Biohippo | Cat# PT-0812/BC-2111 | Refers to viral construct used for stereotaxic injections |
| Transfected construct (*M. musculus*) | PHP.eB-S5E2-C1V1-eYFP | Addgene | Cat# 135633 | Refers to viral construct used for stereotaxic injections |
| Other | Mm-*Drd1*-C1 | Advanced Cell Diagnostics | | RNAscope FISH probe |
| Other | Mm-*Slc1a1*-C2 | Advanced Cell Diagnostics | | RNAscope FISH probe |
| Other | Mm-*Drd2*-C3 | Advanced Cell Diagnostics | | RNAscope FISH probe |
| Other | DAPI stain | Southern Biotech | Cat# 0100–02 | Immunohistochemistry |
| Other | Streptavidin-Alexa Fluor 488 | Jackson Immuno Research | Cat# 016-540-084 | 1:1,000 |
| Other | Streptavidin-Alexa Fluor 647 | Jackson Immuno Research | Cat# 016-600-084 | 1:1,000 |
| Software, algorithm | SPSS v28 | SPSS | | |
| Software, algorithm | IgorPro v6.37 | IgorPro | | |
| Software, algorithm | RStudio 2023.03.1+446 | RStudio | | |

## Mice and genotyping

All mice (*Mus musculus*), males and females, were group housed and kept under a 12 hr light cycle (7:00 AM on, 7:00 PM off) with food and water available ad libitum. Constitutive EAAC1 knockout mice (*Slc1a1$^{-/-}$*) were obtained by targeted disruption of the *Slc1a1* gene, encoding the neuronal glutamate transporter EAAC1, via insertion of a pgk neomycin resistance cassette in exon 1, as originally described by *Peghini et al., 1997*. *Slc1a1$^{-/-}$* breeders were generated after backcrossing *Slc1a1$^{-/-}$* mice with C57BL/6 J wild type mice, here referred to as *Slc1a1$^{+/+}$* (RRID: IMSR_JAX:000664), for >10 generations, as described by *Scimemi et al., 2009*. *Slc1a1$^{+/+}$* breeders from JAX were also bred in house and replaced regularly after 10 generations to avoid genetic drift. *Slc1a1$^{+/+}$* and *Slc1a1$^{-/-}$* mice were identified by PCR analysis of genomic DNA. *Slc1a1$^{tm1c/tm1c}$* mice, here referred to as *Slc1a1$^{f/f}$*, were generated in house from *Slc1a1$^{tm1a(KOMP)Wtsi}$* mice (MGI:4841333) purchased from The Jackson Laboratory. *Gt(ROSA)26Sor$^{tm9/tm9}$* conditional reporter mice, here referred to as *Rosa26$^{tm9/tm9}$* (RRID: IMSR_ JAX:007909; *Madisen et al., 2010*), and *Drd1a$^{tdT/+}$* mice (RRID: IMSR_JAX:016204) were purchased from The Jackson Laboratory. For simplicity, we refer to *Slc1a1$^{tm1c/tm1c}$* and *Slc1a1$^{tm1c/tm1c}$*: *Rosa26$^{tm9/tm9}$* as *Slc1a1$^{f/f}$* mice. *Drd1a$^{Cre/+}$* mice (referred to as *Drd1a$^{Cre/+}$*, RRID: MMRRC_030778-rstudioUCD) and *Adora2a$^{Cre/+}$* mice (referred to as *Adora2a$^{Cre/+}$*, RRID: MMRRC_036158-UCD) (*Gong et al., 2007*; *Gong et al., 2003*) were kindly provided by Dr. Gerfen (NIH/NIDDK). In these mice, the protein Cre-recombinase is expressed under the control of the promoter for D1 receptors (*Drd1a*) and the adenosine receptor 2 (*Adora2a*), respectively. Neurons expressing A2A receptors also express a high density of D2 receptors, and these two receptors establish reciprocal antagonistic interactions in MSNs (*Higley and Sabatini, 2010*). For simplicity, we refer to *Drd1a$^{Cre/+}$*:*Rosa26$^{tm9/tm9}$*, *Adora2a$^{Cre/+}$*:*Rosa26$^{tm9/tm9}$* and *Drd1a$^{tdT/+}$* mice as *Slc1a1$^{+/+}$* mice. Genotyping was performed on toe tissue samples of P7-10 mice. Briefly, tissue samples were digested at 55 °C overnight with shaking at 330 rpm in a lysis buffer containing the following (in mM): 100 Tris base pH 8, 5 EDTA, and 200 NaCl, along with 0.2% SDS and 50 µg/ml proteinase K. Following heat inactivation of proteinase K at 97 °C for 10 min, DNA samples were diluted 1:1 with nuclease-free water. The PCR primers used for *Slc1a1*, *Drd1a$^{Cre/+}$*, *Adora2a$^{Cre/+}$*, *Drd1a$^{tdT/+}$*, and *Rosa26$^{tm9/tm9}$* genotyping were purchased from Thermo Fisher Scientific, and their nucleotide sequences are listed in *Table 1*. PCR was carried out using the KAPA HiFi Hot Start Ready Mix PCR Kit (Cat# KK2602, KAPA Biosystems, Wilmington, MA). Briefly, 12.5 µl of 2 X KAPA HiFi Hot Start Ready Mix was added to 11.5 µl of a diluted primer mix (0.5–0.75 µM final for each primer) and 1 µl of diluted DNA. The PCR cycling protocol for all mutants is described in *Table 2*.

## Trans-cardial perfusion and RNAscope fluorescence in situ hybridzation (FISH)

For RNAscope FISH, mice of either sex aged P30 were anesthetized with a sodium pentobarbital solution (Euthanasia-III Solution, Med-Pharmex, Pomona, CA; 390 mg/ml, 3.9 g/Kg). The mice were perfused through the ascending aorta with 20 ml PBS and 20 ml 4% PFA/PBS at 4 °C at 8 ml/min. The brains were removed, post-fixed with 4% PFA/PBS overnight at 4 °C, cryo-protected in 30% sucrose PBS at 4 °C for 48 hr and stored in PBS for 24 hr. To prepare coronal sections for RNAscope,

**Table 1.** Sequence of genotyping primers.

| Gene | Forward primer | Reverse primer | Band size (bp) |
|---|---|---|---|
| *Slc1a1$^{+/+}$* | 5' ACT CAT CGC AAA CGT CAG TG 3' | 5' GAG AGC AGC AGC CAG TGA TTC 3' | 94 |
| *Slc1a1$^{-/-}$* | 5' CTG TGC TCG ACG TTG TCA CTG 3' | 5' GAG AGC AGC AGC CAG TGA TTC 3' | 680 |
| *Slc1a1$^{+/+}$* | 5' TAC CCC AGT GAC TCA TCA GC 3' | 5' CAT GGT GTT TAC CAG CGT GA 3' | 269 |
| *Slc1a1$^{f/f}$* | 5' TAC CCC AGT GAC TCA TCA GC 3' | 5' CAT GGT GTT TAC CAG CGT GA 3' | 384 |
| *Drd1a$^{Cre/+}$* | 5' GCT ATG GAG ATG CTC CTG ATG GAA 3' | 5' CGG CAA ACG GAC AGA AGC ATT 3' | 340 |
| *Adora2a$^{Cre/+}$* | 5' CGT GAG AAA GCC TTT GGG AAG CT 3' | 5' CGG CAA ACG GAC AGA AGC ATT 3' | 350 |
| *Rosa26$^{-/-}$* | 5' AAG GGA GCT GCA GTG GAG TA 3' | 5' CCG AAA ATC TGT GGG AAG TC 3' | 297 |
| *Rosa26$^{tm9/-}$* | 5' CTG TTC CTG TAC GGC ATG G 3' | 5' GGC ATT AAA GCA GCG TAT CC 3' | 196 |
| *Drd1a$^{tdT/+}$* | 5' CTT CTG AGG CGG AAA GAA CC 3' | 5' TTT CTG ATT GAG AGC ATT CG 3' | 750 |

**Table 2.** PCR protocol for all mice.

|  | Initiation/melting | Denaturation | Annealing | Elongation | Amplification | Hold |
|---|---|---|---|---|---|---|
| Temperature | 95 °C | 95 °C | 60 °C | 72 °C | 72 °C | 4 °C |
| Duration | 3 min | 0.25 min | 0.25 min | 0.25 min | 1 min | ∞ |
| Cycles | 1 | 35 |  |  | 1 |  |

we separated the two hemispheres, embedded them in 4% w/v agar/PBS and prepared 40-μm-thick slices using a vibrating blade microtome (VT1200S, Leica Microsystems, Buffalo Grove, IL). The slices were post-fixed in 4% PFA/PBS for 30 min at room temperature (RT) and mounted onto Superfrost plus microscope slides. Mounted slices were used for fluorescence in situ hybridization (FISH) using an RNAscope multiplex fluorescent assay (Advanced Cell Diagnostics, Newark, CA) according to manufacturer instructions, using Mm-Drd1-C1, Mm-Sc1a1-C2 and Mm-Drd2-C3 RNA probes and Opal 520, 570 and 690 dyes (Akoya Biosciences, Menlo Park, CA). DAPI Fluoromount G was used as the mounting medium (Cat# 0100–02; SouthernBiotech, Birmingham, AL). The presence of mRNA transcripts was assessed using a confocal microscope (Zeiss LSM710) equipped with a Plan-Apochromat 63 X/1.4NA oil objective. Image size was set to 1,024 · 1,024 pixels and represented the average of 8 consecutive frames. The image analysis was performed using CellProfiler version 4.0.7 and a modified version of the Speckle Counting Pipeline (*Carpenter et al., 2006*). In this pipeline, we set a feature size of 20 for the EnhanceOrSuppressFeatures tool. We used Advanced Settings for the IdentifyPrimaryObjects tool. Here, the typical diameter of objects was set to 3–20 pixels, the selected thresholding method was Robust Background, and the Upper Outlier Fraction was set to 0. The size of the smoothing filter for declumping was set to 4. The minimal allowed distance to suppress local maxima was set to 4. We did not use lower-resolution images to find local maxima. The EnhanceOrSuppressFeatures, MaskImage, IdentifyPrimaryObjects and RelateObjects tools were applied for each label color (green, red, far red; i.e. three times).

## Stereotaxic intracranial injections and optogenetics

AAV2/5-EF1a-DIO-hChR2(H134R)-EYFP-WPRE-pA (University of North Carolina Vector Core, Chapel Hill, NC) was injected into *Drd1a*^Cre/+, *Adora2a*^Cre/+, *Drd1a*^Cre/+:*Slc1a1*^-/-, *Adora2a*^Cre/+:*Slc1a1*^-/- mice of either sex aged P14-16. AAV2/9-D1-Cre-EGFP-WPRE-hGH-pA (Biohippo cat# PT-0812/BC-2111, Gaithersburg, MD; here referred to as AAV-D1Cre) was injected into the DLS, VMS or SN/VTA of *Slc1a1*^f/f mice of either sex aged P14-16. PHP.eB-S5E2-C1V1-eYFP (Addgene cat# 135633) was injected in *Drd1a*^Cre/+:*Rosa26*^tm9/tm9 mice of either sex aged P14-16. This virus transfected ~2% of all cells in the DLS. For the stereotaxic injections, mice were anesthetized with isoflurane (induction: 5% in 100% O$_2$ at 1–2 l/min; maintenance: 3% in 100% O$_2$ at 1–2 l/min) and placed in the stereotaxic frame of a motorized drill and injection robot (Neurostar GmbH, Tübingen, Germany). After making a skin incision and thinning the skull under aseptic conditions, we injected 200 nl of the viral constructs bilaterally in the DLS using a Hamilton syringe at a rate of 50 nl/min. The injection coordinates from bregma were AP:+0.2 mm, ML:±2.2 mm, DV: 2.5 mm for the DLS; AP:+0.4 mm, ML:±2.0 mm, DV: 3.7 mm for the VMS; AP: –3.1 mm, ML:±1.2 mm, DV: 4.3 mm for the SN/VTA. After the stereotaxic injections, the mice were returned to their home cage and used for slice physiology experiments 2–3 weeks after surgery.

## Acute slice preparation

Acute coronal slices of the mouse striatum were obtained from *Drd1a*^Cre/+:*Rosa26*^tm9/tm9, *Adora2a*^Cre/+:*Rosa26*^tm9/tm9, *Drd1a*^tdT/+ and *Drd1a*^Cre/+:*Rosa26*^tm9/tm9:*Slc1a1*^-/-, *Adora2a*^Cre/+:*Rosa26*^tm9/tm9: *Slc1a1*^-/-, *Drd1a*^tdT/+:*Slc1a1*^-/- mice of either sex (P28-37), deeply anesthetized with isoflurane and decapitated in accordance with SUNY Albany Animal Care and Use Committee guidelines. The brain was rapidly removed and placed in ice-cold slicing solution bubbled with 95% O$_2$/5% CO$_2$ containing the following (in mM): 119 NaCl, 2.5 KCl, 0.5 CaCl$_2$, 1.3 MgSO$_4$·H$_2$O, 4 MgCl$_2$, 26.2 NaHCO$_3$, 1 NaH$_2$PO$_4$, and 22 glucose, 320 mOsm, pH7.4. The slices (250 μm thick) were prepared using a vibrating blade microtome (VT1200S; Leica Microsystems, Buffalo Grove, IL). Once prepared, the slices were stored in this solution in a submersion chamber at 36 °C for 30 min and at RT for at least 30 min and up to 5 hr.

## Electrophysiology, optogenetics, and glutamate uncaging

Unless otherwise stated, the recording solution contained the following (in mM): 119 NaCl, 2.5 KCl, 1.2 $CaCl_2$, 1 $MgCl_2$, 26.2 $NaHCO_3$, and 1 $NaH_2PO_4$, 22 glucose, 300 mOsm, pH7.4. In experiments performed in the absence of external $Mg^{2+}$, the $CaCl_2$ concentration was increased to 3.8 mM. We identified the DLS under bright field illumination using an upright fixed-stage microscope (BX51 WI; Olympus, Center Valley, PA). We raised the extracellular $CaCl_2$ concentration to 4 mM to record mEPSCs and mIPSCs. When recording evoked IPSCs, stimulating and recording electrodes were both placed in the DLS ~100 μm away from each other. Recordings from D1- and D2-MSNs, visually identified for their expression of the reporter protein tdTomato, were made with patch pipettes containing (in mM): 120 $CsCH_3SO_3$, 10 EGTA, 20 HEPES, 2 MgATP, 0.2 NaGTP, 5 QX-314Br, 290 mOsm, pH 7.2. Postsynaptic responses were evoked using electrical or optogenetic stimulation. Electrical stimulation was obtained by delivering constant voltage electrical pulses (50 μs) through a stimulating bipolar stainless-steel electrode (Cat# MX21AES(JD3); Frederick Haer Corporation, Bowdoin, ME). To activate ChR2-expressing fibers from MSNs or C1V1-expressing PV-INs, we used 5 ms-long light pulses generated by a SOLA-SE light engine (Lumencor, Beaverton, OR) and filtered using either a green FITC filter set (469/497/525) or a red TRITC filter set (542/570/620). The light power at the sample plane was ~250 μW and the light pulses were delivered at intervals of 30 s using whole-field illumination through a 40 X water immersion objective (LUMPLFLN40XW; Olympus, Center Valley, PA). For all electrophysiology experiments, the resistance of the recording electrode was 5–7 MOhm and was monitored throughout the experiments. Data were discarded if the resistance changed >20% during the experiment. When recording excitatory currents, picrotoxin (100 μM) was added to the recording solution to block $GABA_A$ receptors. All recordings were obtained using a Multiclamp 700B amplifier and filtered at 5 KHz (Molecular Devices, San Jose, CA), converted with an 18-bit 200 kHz A/D board (HEKA, Holliston, MA), digitized at 10 KHz, and analyzed offline with custom-made software (A.S.) written in IgorPro 6.37 (Wavemetrics, Lake Oswego, OR; available at https://github.com/scimemia/EAAC1-lateral-inhibition, *Scimemia, 2023*). Tetrodotoxin (TTX; Cat# T-550) was purchased from Alomone Labs (Jerusalem, Israel). NBQX disodium salt (Cat# HB0443) and D,L-APV (Cat# HB0251), were purchased from Hello Bio (Princeton, NJ). (3 S)–3-[[3-[[4-(Trifluoromethyl)benzoyl]amino]phenyl]methoxy]-L-aspartic acid, here referred to as T-TBOA (Cat# 2532), was purchased from Tocris Bio-Techne (Minneapolis, MN). D-2-Aminoadipic acid, 98% (D-AA; Cat# AAH27345MD) was purchased from Alfa Aesar (Tewksbury, MA). All other chemicals were purchased from Millipore Sigma. All recordings were performed at RT. Electrophysiological recordings were analyzed within the IgorPro 6.37 environment using custom made software (A.S.).

## Confocal imaging of biocytin filled MSNs and spine analysis

Biocytin 0.2%–0.4% (w/v) was added to the intracellular solution used to patch MSNs. Each cell was filled for at least 20 min. The slices were then fixed overnight at 4 °C in 4% PFA/PBS, cryo-protected in 30% sucrose PBS, and incubated in 0.1% streptavidin-Alexa Fluor 488 or streptavidin-Alexa Fluor 647 conjugate and 0.1% Triton X-100 for 3 hr at RT. The slices were then mounted onto microscope slides using Fluoromount-G or DAPI Fluoromount-G mounting medium (Cat# 0100–01 and Cat# 0100–02, respectively; SouthernBiotech, Birmingham, AL). Confocal images were acquired using a Zeiss LSM710 inverted microscope equipped with 488 nm Ar or 633 nm HeNe laser. All images were acquired as stitched z-stacks of 4–5 frames averages (1024x1024 pixels; 1 μm z-step) using a 40 X/1.4 NA or 63 X/1.4NA Plan-Apochromat oil-immersion objectives. We generated 2D maximum intensity projections of biocytin-filled MSNs and traced the dendritic processes manually using Simple Neurite Traces in Fiji (https://fiji.sc/). Confocal images for spine analysis were also collected as z-stacks of 8 frame averages (1024x1024 pixels; 0.5 μm z-step; 3–5 digital zoom) using a 63 X/1.4 NA Plan-Apochromat oil-immersion objective. For structural analysis, dendritic spines were classified into four groups according to their neck and head size (i.e. mushroom, thin, stubby, filopodia), using Fiji (*Peters and Kaiserman-Abramof, 1970*; *Jones and Powell, 1969*; *Harris et al., 1992*).

## NEURON model of D1-MSNs: 3D realistic morphologies of D1-MSNs

All files pertaining to the 3D realistic morphology model and simulation were uploaded to the ModelDB database (https://senselab.med.yale.edu/modeldb/, ModelDB acc.n. 267267, "Realistic Morphology" folder). 3D reconstructions of biocytin-filled D1-MSNs from *Slc1a1*$^{+/+}$ and *Slc1a1*$^{-/-}$ mice

were created using the SNT plugin for Fiji (https://fiji.sc/). The dendritic arbor of these cells differed between *Slc1a1*$^{+/+}$ and *Slc1a1*$^{-/-}$ mice, consistent with data collected from 3D reconstructions of biocytin-filled D1-MSNs (*Figure 2*). These morphologies were used to create a NEURON model that was populated with voltage-gated sodium, calcium (Ca$_V$1.2, Ca$_V$1.3), and potassium channels (K$_{AS}$, K$_{DR}$, K$_{IR}$, K$_{RP}$). To closely reproduce the main passive and active membrane properties of D1-MSNs measured experimentally, including their resting membrane potential, action potential threshold and peak (*Figure 7—figure supplement 1A–D*), and repetitive firing rate in response to sustained (500ms) somatic current injections of increasing amplitude (10–80 pA; *Figure 7—figure supplement 1E–G*). Each conductance was adjusted as follows: sodium channels (2.4 S/cm$^2$); voltage-gated calcium channels (Ca$_V$1.2: 6.7e-6 S/cm$^2$, Ca$_V$1.3: 1.0e-4 S/cm$^2$) and potassium channels (K$_{AS}$: 4.0e-5 S/cm$^2$, K$_{DR}$: 5.0e-3 S/cm$^2$, K$_{IR}$: 1.0e-4 S/cm$^2$, K$_{RP}$: 2.0e-4 S/cm$^2$). Voltage-gated sodium channels were placed in the axon initial segment (a 50 µm axonal segment with a diameter of 0.5 µm). All voltage-gated calcium and potassium channels were distributed throughout the plasma membrane of the soma and the dendrites. Passive leak channels were present throughout the plasma membrane.

We did not make assumptions on the location of the synaptic inputs active during our electrophysiology recordings, or on the potential existence of synaptic scaling mechanisms that might balance for passive attenuation of distal synaptic inputs, as this has not been documented for D1-MSNs (*Nicholson et al., 2006*; *Magee and Cook, 2000*). Instead, we assumed that all inputs onto D1-MSNs are of equal strength, regardless of which portion of the dendrite they target. This assumption was important to set the synaptic weights of E/I inputs to ensure the model reproduced the amplitude of mEPSCs and mIPSCS recorded experimentally from D1-MSNs. To accomplish this, each simulation was repeated 100 times, each time randomizing the location of either one excitatory or inhibitory input (*Figure 7—figure supplement 2A*, *Figure 7—figure supplement 4A*). We then averaged the somatic mPSCs and compared their amplitude and kinetics with those of mPSCs recorded from D1-MSNs in *Slc1a1*$^{+/+}$ mice. We performed an initial adjustment of the peak conductance of the synaptic inputs in the model (AMPA: 7.2e-4 µS, GABA$_A$: 3.6e-4 µS) to match the mPSC amplitude recorded from experiments.

We modeled mPSCs decay using a mono-exponential function, whereby $decay = A_1 e^{-t/\tau_1}$ . We set the amplitude (A$_1$) and decay time constant ($\tau_1$) of mPSCs in the model to match that of A$_1$ and $\tau_1$ measured across experiments (*Figure 7—figure supplement 2C, D*, *Figure 7—figure supplement 4C, D*). We tested a range of combinations of A$_1$ and $\tau_1$ that could reproduce either the mPSC amplitude or decay time (*Figure 7—figure supplement 2B*; *Figure 7—figure supplement 4B*), to find the one that would accurately reproduce both the amplitude and kinetics of mPSCs recorded experimentally in D1-MSNs in *Slc1a1*$^{+/+}$ and *Slc1a1*$^{-/-}$ mice (*Figure 7—figure supplement 2E, F*; *Figure 7—figure supplement 4E, F*). In the case of NMDA EPSCs, we did not make a direct comparison between the model and experimental NMDA mEPSCs, as these events are particularly small in D1-MSNs, and technically challenging to record. Instead, we relied on knowledge of the AMPA mEPSC amplitude at V$_{hold}$=-70 mV (*Figure 3*), the reversal potential for glutamatergic currents (0 mV), and the experimentally measured value for the NMDA/AMPA ratio corrected by the driving force of each current (*Slc1a1*$^{-/-}$ 0.57±0.14; *Slc1a1*$^{+/+}$ 0.68±0.14; p=0.57; *Figure 3—figure supplement 1A, B*), to predict the NMDA mEPSC amplitude at V$_{hold}$ = 40 mV. The estimated NMDA mEPSC amplitude was calculated as the product of the AMPA mEPSC amplitude at V$_{hold}$=-70 mV, the NMDA/AMPA ratio, and the ratio of the NMDA and AMPA driving force (*Slc1a1*$^{+/+}$ 15.0 pA · 0.68 · 40 mV/70 mV =5.8 pA; NMDA mEPSC *Slc1a1*$^{-/-}$ 19.3 pA · 0.57 · 40 mV/70 mV =6.3 pA). This estimated mEPSC amplitude was used for the initial adjustment of the peak conductance of the synaptic NMDA input in the model (NMDA: 6.8e-5 µS). The estimated NMDA mEPSC decay matched that of evoked NMDA EPSCs recorded experimentally (*Figure 7—figure supplement 3*). We then determined the values for A$_1$ and $\tau_1$ for synaptic NMDA inputs in our model (*Figure 7—figure supplement 3C, D*) using the same method previously mentioned for synaptic AMPA and GABA inputs. Together, these parameter-constraints allowed the compartmental model to reproduce the larger AMPA mEPSCs amplitude (*Figure 3A–D*), similar NMDA/AMPA ratio (*Figure 3—figure supplement 1*), and smaller GABA mIPSC amplitude and decay time in D1-MSNs from *Slc1a1*$^{-/-}$ mice (*Figure 4*).

The models did not account for differences in tonic inhibition between D1-MSNs of *Slc1a1*$^{+/+}$ and *Slc1a1*$^{-/-}$ mice, because we verified experimentally that the tonic GABA currents in these cells are similar (*Slc1a1*$^{+/+}$: 4.0±1.4 pA; *Slc1a1*$^{-/-}$: 5.8±5.9 pA; *P*=0.78). These currents were measured as the picrotoxin-sensitive currents measured in cells held at –70 mV. Similar results were obtained when

comparing the tonic current density, obtained by dividing the tonic currents by the capacitance of each cell ($Slc1a1^{+/+}$: 0.056±0.019 pA/pF; $Slc1a1^{-/-}$: 0.052±0.081 pA/pF; $P$=0.97).

## Behavioral analysis

We performed our behavioral analysis on C57BL/6 J wild type and $Slc1a1^{-/-}$ mice of either sex aged P28-37, maintained on as 12 h:12 h L/D cycle. Three days prior to training, and for the whole duration of the training and testing sessions, the drinking water in the home cage was supplemented with 2% citric acid (w/v) (*Reinagel, 2018*; *Whiteway et al., 2021*). Each behavioral session lasted 5 min. During each one of the ten training sessions, a 70 µl water reward was delivered in response to a lever press, and the timing of each lever press and reward was monitored using custom made MATLAB code and a B-pod state machine (Sanworks, Rochester, NY). The reward probability was set to $P_{rew}$ = 0.5∥0.9. Each mouse was subject to 10 training sessions with $P_{rew}$ = 0.5 and 10 training sessions with $P_{rew}$ = 0.9. Mice that did not perform lever presses during this training period were excluded from further analysis. During the one final testing session, we changed the reward probability from $P_{rew}$ = 0.5 to $P_{rew}$ = 0.9 at one of the following switching time intervals (5, 10, 15, 30, 50, 75 s). The number and time distribution of lever presses and rewards for each mouse was analyzed in IgorPro 6.37 (Wavemetrics, Lake Oswego, OR).

## Quantification and statistical analysis

All experiments were conducted blind with regard to mouse genotype. Sample size determination was based on power analysis. Data averages are presented as mean ± SEM, unless indicated otherwise. PCA and t-SNE analysis was conducted in RStudio, and the graphical representations relied on the use of the ggplot2 package. Statistical analysis was performed using IgorPro 6.37 or IBM SPSS Statistics 28. Statistical significance was determined by Student's paired or unpaired t-test as appropriate or, when comparing multiple groups, by one- or two-way ANOVA, with or without repeated measures. F-statistics and p-values for ANOVA tests that did not reach statistical significance and were not followed by *post hoc* t-tests are included in the text. Statistically significant ANOVA tests were followed by *post hoc* t-test comparisons (reported in the text). A linear correlation test was used to compare the Sholl analysis results in *Figure 2*. The corresponding r-value is reporter in the text. A full report of all statistical comparisons for this manuscript is included in the data sheets shared via Open Science Framework. Differences were considered significant at p<0.05 (*p<0.05; **p<0.01; ***p<0.001).

## Acknowledgements

We thank Dr. Bernardo L Sabatini for comments on the manuscript and valuable discussions, and Drs. Daniela Tropea, Paolo Guasoni and Damian G Zuloaga for technical advice on statistical analysis.

## Additional information

### Funding

| Funder | Grant reference number | Author |
| --- | --- | --- |
| National Science Foundation | IOS1655365 | Annalisa Scimemi |
| National Science Foundation | IOS2011998 | Annalisa Scimemi |

The funders had no role in study design, data collection and interpretation, or the decision to submit the work for publication.

### Author contributions

Maurice A Petroccione, Data curation, Formal analysis, Investigation, Visualization; Lianna Y D'Brant, Formal analysis, Investigation; Nurat Affinnih, Patrick H Wehrle, Gabrielle C Todd, Shergil Zahid, Haley E Chesbro, Ian L Tschang, Investigation; Annalisa Scimemi, Conceptualization, Resources, Data

curation, Software, Formal analysis, Supervision, Funding acquisition, Validation, Investigation, Visualization, Methodology, Writing - original draft, Project administration, Writing - review and editing

### Author ORCIDs
Maurice A Petroccione http://orcid.org/0000-0003-4333-4310
Lianna Y D'Brant https://orcid.org/0000-0001-7181-574X
Nurat Affinnih http://orcid.org/0000-0002-4203-6410
Patrick H Wehrle https://orcid.org/0000-0002-8732-2699
Gabrielle C Todd https://orcid.org/0000-0002-9771-2623
Shergil Zahid https://orcid.org/0000-0001-7628-8811
Haley E Chesbro https://orcid.org/0000-0002-3421-8118
Ian L Tschang https://orcid.org/0000-0002-7599-1668
Annalisa Scimemi http://orcid.org/0000-0003-4975-093X

### Ethics
All experimental procedures were performed in accordance with protocols approved by the Institutional Animal Care and Use Committee at the State University of New York (SUNY) Albany and guidelines described in the National Institutes of Health's Guide for the Care and Use of Laboratory Animals.

### Decision letter and Author response
Decision letter https://doi.org/10.7554/eLife.81830.sa1
Author response https://doi.org/10.7554/eLife.81830.sa2

## Additional files

### Supplementary files
• MDAR checklist

### Data availability
All primary data used in this work and complete statistical analyses for each figure have been deposited to the Open Science Framework (https://osf.io/dw5n7/).

The following dataset was generated:

| Author(s) | Year | Dataset title | Dataset URL | Database and Identifier |
|---|---|---|---|---|
| Scimemi | 2023 | Neuronal glutamate transporters control reciprocal inhibition and gain modulation in D1 medium spiny neurons | https://osf.io/dw5n7/ | Open Science Framework, dw5n7 |

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
