## [Editor Report]

This fundamental study outlines the role of the neuronal glutamate transporter (EAAC1) in striatal function and behavior. The evidence supporting the conclusions is compelling, with rigorous studies spanning genetic approaches, physiology, and behavior. This work will be of general interest to those studying striatal biology as well as the neural control of behavior.

---

## [Decision Letter]

**Decision letter after peer review:**

Thank you for submitting your article "Neuronal glutamate transporters control reciprocal inhibition and gain modulation in D1 medium spiny neurons" for consideration by *eLife*. Your article has been reviewed by 2 peer reviewers, and the evaluation has been overseen by a Reviewing Editor and Lu Chen as the Senior Editor. The reviewers have opted to remain anonymous.

Essential revisions:

The reviewers were in agreement that the manuscript has exciting data, and the concerns were similar across all of the reviewers.

The overarching concerns that were agreed upon by the reviewers and editors were as follows:

1. Reviewer 1's point about support for the mechanism of the change in IPSCs. The authors either need to consider additional experiments to support the claims that were made or tone down the discussion and remove the cartoon of the proposed mechanism.

2. The concerns that would require the most consideration is focused on those of regional specificity. For claims of regional specificity in the DLS, the authors would need to do region-specific manipulations. Because of the use of a constitutive knockout mouse these claims cannot be supported by the data at hand.

3. Considering the OCD framework within which the manuscript is presented since these data are more broadly applicable to compulsion in general rather than OCD as a disorder.

In the discussion of this manuscript it was agreed that these three concerns must be addressed for further consideration; however, there were other concerns that also should be addressed as listed below in each reviewer's individual summary.

*Reviewer #1 (Recommendations for the authors):*

Figure 4/5/6. The authors report effects of TBOA on mIPSCs, evoked and oIPSCs in D1 and D2 cells that they attribute to specific effects of EAAC1 on GABA synthesis/supply. While the conclusions related to the EAAC1-regulated changes in inhibitory transmission are clear, I am not convinced that it can be entirely attributed to GABA supply. Beyond assessing the PPR, the authors have not performed experiments testing the cartoon shown in Figure 4A, especially considering the non-mutually exclusive possibility that mGluR/CB1R mediated suppression of GABA release could also be a possibility (i.e. Drew et al., J Neurosci 2008 and refs within, noting this mechanism occurs in several brain regions), especially considering that EAAC1 limits mGluR activation (Bellini 2018). I don't think that the authors need to dissect the mechanism for reduced IPSCs for the current results to be interesting and important, but a few straightforward pharmacology experiments would be informative. But without additional supportive experiments, the cartoon and estimates of %GABA release could be removed, and discussed as one of multiple mechanisms that could suppress IPSCs in EAAC1 KO cells.

One of the most interesting results is the robust effect on D1 morphology, but there was little discussion of this unexpected finding, and why it would only occur in D1 and not D2 MSNs. Has neuronal morphology been assessed in other populations after deletion of EAAC1 (or other glutamate transporters)?

*Reviewer #2 (Recommendations for the authors):*

1. There are several experimental paradigms that could be done to link the physiology with the behavior in a stronger way (any of which would be acceptable):

a. Microinfusions of EAAC1 antagonist into DLS.

b. Cre-dependent CRISPR virus or shRNA directed toward EAAC1 in D1-Cre mouse.

c. Conditional deletion of EAAC1.

d. Manipulations that may recapitulate function of EAAC1 – increase GABAergic synaptic strength at axon collaterals (Cre-dependent axonal Gq-DREADD in D1-Cre mouse w/ CNO microinfusion).

e. Local rescue of EAAC1 in EAAC1KO mouse using HSV or Lenti containing EAAC1 injected into DLS.

---

## [Author Response]

Essential revisions:The reviewers were in agreement that the manuscript has exciting data, and the concerns were similar across all of the reviewers.

We would like to thank the editor and reviewers for the thorough and thoughtful evaluation of the manuscript. We have addressed the concerns that were raised during the evaluation process, which provided us with an opportunity to strengthen the main findings of this work. We provide a succinct answer to the essential revisions below, together with more detailed and point-to-point answers to the specific concerns raised by each reviewer.

The overarching concerns that were agreed upon by the reviewers and editors were as follows:1. Reviewer 1's point about support for the mechanism of the change in IPSCs. The authors either need to consider additional experiments to support the claims that were made or tone down the discussion and remove the cartoon of the proposed mechanism.

We addressed this concern by adding new sets of pharmacology experiments. The results showed that the reduction in the amplitude of GABAergic currents (mIPSCs and evoked IPSCs) in D1-MSNs induced by the glutamate transporter blocker T-TBOA persists when T-TBOA is applied in the continued presence of CB1 and/or mGluRI-III antagonists (Supp. Figure 4-6). These new findings indicate that the ability of EAAC1 to increase action potential-dependent and -independent GABA release is not due to activation of CB1 and/or mGluRI-III receptors. The findings are now included in the revised version of the manuscript. Due to the outcome of these experiments, the cartoon of the proposed mechanism of action of EAAC1 has been included as part of the manuscript figures.

2. The concerns that would require the most consideration is focused on those of regional specificity. For claims of regional specificity in the DLS, the authors would need to do region-specific manipulations. Because of the use of a constitutive knockout mouse these claims cannot be supported by the data at hand.

This is an important point, which we addressed by repeating our behavioral assay (reward-based task switching) in three different experimental conditions.

First, we used conditional EAAC1 knockout mice lacking EAAC1 in either D1- or D2-MSNs (D1^Cre/+^:EAAC1^f/f^ and A2A^Cre/+^:EAAC1^f/f^, respectively). The fact that mice lacking EAAC1 in D1-MSNs show a similar task-switching behavior compared to that of EAAC1^-/-^ mice, whereas mice lacking EAAC1 in D2-MSNs behave similarly to EAAC1^+/+^ mice is consistent with the hypothesis that EAAC1 controls the function of neurons implicated in the execution of switching tasks by regulating excitation and homosynaptic lateral inhibition in D1-MSNs (Supp. Figure 8-1).

Second, we performed stereotaxic injections of a Cre-dependent virus (AAV-D1Cre) in the dorsolateral (DLS) or ventromedial striatum (VMS) of EAAC1^f/f^ mice, to remove EAAC1 expression from D1-MSNs in these striatal domains (Supp. Figure 8-2). The results of these experiments showed that the task-switching behavior of these mice is indistinguishable from that of EAAC1^-/-^ mice, but different from that of EAAC1^+/+^ mice. This suggests that excitation and lateral inhibition of D1-MSNs in the DLS and the VMS are implicated with the execution of reward-based task switching behaviors. We revised the text of our manuscript accordingly.

Third, since D1-MSNs are also abundantly expressed in the SN/VTA, we performed stereotaxic injections of AAVD1Cre in the SN/VTA of EAAC1^f/f^ mice, to remove EAAC1 expression in these cells and determine whether there is a regional specificity of EAAC1 in striatal D1-MSNs (Supp. Figure 8-3). The experiments showed that the task switching behavior of these mice was indistinguishable from that of EAAC1^+/+^ mice and differed from that of EAAC1^-/-^ mice, suggesting that EAAC1 alters task-switching behaviors by altering the activity striatal D1-MSNs, not of other D1-MSNs like those present in the SN/VTA. These results are described and discussed in the revised version of the manuscript.

3. Considering the OCD framework within which the manuscript is presented since these data are more broadly applicable to compulsion in general rather than OCD as a disorder.

We agree with the reviewers and have toned down the OCD framework that was originally included in the first submission of this manuscript. The revised text discusses our data in the broader context of compulsions, as suggested by the reviewers.

In the discussion of this manuscript it was agreed that these three concerns must be addressed for further consideration; however, there were other concerns that also should be addressed as listed below in each reviewer's individual summary.

We have now addressed all the concerns raised by all reviewers.

Reviewer #1 (Recommendations for the authors):Figure 4/5/6. The authors report effects of TBOA on mIPSCs, evoked and oIPSCs in D1 and D2 cells that they attribute to specific effects of EAAC1 on GABA synthesis/supply. While the conclusions related to the EAAC1-regulated changes in inhibitory transmission are clear, I am not convinced that it can be entirely attributed to GABA supply. Beyond assessing the PPR, the authors have not performed experiments testing the cartoon shown in Figure 4A, especially considering the non-mutually exclusive possibility that mGluR/CB1R mediated suppression of GABA release could also be a possibility (i.e. Drew et al., J Neurosci 2008 and refs within, noting this mechanism occurs in several brain regions), especially considering that EAAC1 limits mGluR activation (Bellini 2018). I don't think that the authors need to dissect the mechanism for reduced IPSCs for the current results to be interesting and important, but a few straightforward pharmacology experiments would be informative. But without additional supportive experiments, the cartoon and estimates of %GABA release could be removed, and discussed as one of multiple mechanisms that could suppress IPSCs in EAAC1 KO cells.

We agree with the reviewers and addressed this concern by measuring the effect of T-TBOA on mIPSCs (Supp. Figure 4-1) and IPSCs (Supp. Figure 5-1) recorded in the continued presence of CB1 and/or mGluRI-III antagonists. The results of these experiments indicate that the effect of T-TBOA is similar in the presence or absence of these drugs, supporting the hypothesis that EAAC1 contributes to GABA synthesis and release. These new data are now included and discussed in the revised version of this manuscript.

One of the most interesting results is the robust effect on D1 morphology, but there was little discussion of this unexpected finding, and why it would only occur in D1 and not D2 MSNs. Has neuronal morphology been assessed in other populations after deletion of EAAC1 (or other glutamate transporters)?

We addressed this concern by adding a section at the beginning of the *Discussion* in the revised manuscript. Briefly, we are not aware of other works showing that genetic deletion of EAAC1 leads to changes in neuronal morphology. In fact, our own past analysis in the hippocampus indicated that the spine density and spine head diameter are similar in CA1 pyramidal cells of EAAC1^+/+^ and EAAC1^-/-^ mice (Scimemi, Tian, and Diamond 2009). It is worth mentioning, however, that this previous work did not include a Sholl analysis or a detailed analysis of spine type distribution, like the one included in the current manuscript. It remains unknown whether morphological changes similar to those detected in D1-MSNs of EAAC1^-/-^ mice can be induced by inducing a similar change in the expression of glial glutamate transporters. Although there is evidence that the expression of glial glutamate transporters is altered in neuropsychiatric disorders associated with a reduction in spine density and dendritic arbors (O'Donovan, Sullivan, and McCullumsmith 2017), a causal link between these effects remains to be established. It might be tempting to speculate that the increase in spine size and dendritic complexity in D1MSNs of EAAC1^-/-^ mice might be due to increased spillover, since there is evidence that glutamate triggers morphological changes in neurons and promotes the acquisition of a mature spine morphology largely mediated by NMDA receptors (Mattison et al. 2014; Kwon and Sabatini 2011). If this were true, one could hypothesize that the lack of effect on D2-MSNs might be due to a lack of effect of EAAC1 on synaptic transmission onto these cells. This, however, would not be supported by the results obtained in the hippocampus, where EAAC1 limits spillover yet has no detectable effect on spine size (Scimemi, Tian, and Diamond 2009). We also briefly discuss the possibility that these may be indirect and mediated by dopamine. While these findings indicate that there might multiple and complex signaling pathways implicated in the control of spine and dendrite size, they also point to EAAC1 as a critical component of these regulatory mechanisms.

Reviewer #2 (Recommendations for the authors):1. There are several experimental paradigms that could be done to link the physiology with the behavior in a stronger way (any of which would be acceptable):a. Microinfusions of EAAC1 antagonist into DLS.b. Cre-dependent CRISPR virus or shRNA directed toward EAAC1 in D1-Cre mouse.c. Conditional deletion of EAAC1.d. Manipulations that may recapitulate function of EAAC1 – increase GABAergic synaptic strength at axon collaterals (Cre-dependent axonal Gq-DREADD in D1-Cre mouse w/ CNO microinfusion).e. Local rescue of EAAC1 in EAAC1KO mouse using HSV or Lenti containing EAAC1 injected into DLS.

We appreciate the recommendations and constructive criticism provided by the reviewer. The pharmacology of EAAC1 is rather complex, and unfortunately to date there are no proven specific EAAC1 antagonists, validated against EAAC1^-/-^ mice, that do not cross-react also with glial glutamate transporters. However, as noted by the reviewer, there are genetic and viral approaches that allow us to overcome the technical limitations of pharmacology and that can be used to test the cell- and region-specificity of the effects of EAAC1 on behavior. Our approach relied on the following strategies.

*First,* we performed our task switching behavioral tests in conditional mice lacking EAAC1 in either D1- or D2MSNs (D1^Cre/+^:EAAC1^f/f^ and A2A^Cre/+^:EAAC1^f/f^ mice, respectively; Supp. Figure 8-1). The results showed that the behavior of D1^Cre/+^:EAAC1^f/f^ mice is similar to that of EAAC1^-/-^ mice, whereas that of A2A^Cre/+^:EAAC1^f/f^ mice is similar to that of EAAC1^+/+^ mice.

*Second,* we injected a Cre-dependent AAV (AAV-D1Cre) in the DLS or VMS of EAAC1^f/f^ mice, to remove EAAC1 expression from D1-MSNs in either the DLS or VMS (Supp. Figure 8-2). The results of these experiments showed that mice devoid of EAAC1 in DLS or VMS D1-MSNs show similar behaviors compared to EAAC1^-/-^ mice, suggesting that the expression of EAAC1 in D1-MSNs of the DLS and VMS D1-MSNs are both implicated with the execution of task switching behaviors.

*Third*, we asked whether removing EAAC1 from D1 receptor-expressing neurons in other basal ganglia nuclei could also mimic the behavior of constitutive EAAC1^-/-^ mice and D1^Cre/+^:EAAC1^f/f^ mice. To accomplish this, we injected AAV-D1Cre in the SN/VTA of EAAC1^f/f^ mice (Supp. Figure 8-3). The results showed that the task switching behavior of these mice was indistinguishable from that of EAAC1^+/+^ mice, and different from that of EAAC1^-/-^ and D1^Cre/+^:EAAC1^f/f^ mice. This finding suggests that EAAC1 controls task-switching behaviors by modulating excitation and homosynaptic lateral inhibition in D1-MSNs of the striatum (DLS and VMS; Supp. Figure 8-2) and not in D1-MSNs present the SN/VTA.

Although we considered additional strategies, like the axon-specific DREADD approach mentioned by the reviewer, we realized that these will not recapitulate the increased GABAergic inhibition at D1-D1 synapses detected in EAAC1^-/-^ mice but would rather lead to a global increase in GABA release at all synapses formed by D1-MSNs, including those formed onto other cells in and out of the DLS (e.g., D2-MSNs, neurons in the SNc and GPE). For these reasons, this approach was not pursued further.

References

Barbour, B., B. U. Keller, I. Llano, and A. Marty. 1994. 'Prolonged presence of glutamate during excitatory synaptic transmission to cerebellar Purkinje cells', *Neuron*, 12: 1331-43.

Boyson, S. J., P. McGonigle, and P. B. Molinoff. 1986. 'Quantitative autoradiographic localization of the D1 and D2 subtypes of dopamine receptors in rat brain', *J Neurosci*, 6: 3177-88.

Cadet, J. L., S. Jayanthi, M. T. McCoy, G. Beauvais, and N. S. Cai. 2010. 'Dopamine D1 receptors, regulation of gene expression in the brain, and neurodegeneration', *CNS Neurol Disord Drug Targets*, 9: 526-38.

Diamond, J. S., and C. E. Jahr. 1995. 'Asynchronous release of synaptic vesicles determines the time course of the AMPA receptor-mediated EPSC', *Neuron*, 15: 1097-107.

Gu, B. M., J. Y. Park, D. H. Kang, S. J. Lee, S. Y. Yoo, H. J. Jo, C. H. Choi, J. M. Lee, and J. S. Kwon. 2008. 'Neural correlates of cognitive inflexibility during task-switching in obsessive-compulsive disorder', *Brain*, 131: 155-64.

Hestrin, S., P. Sah, and R. A. Nicoll. 1990. 'Mechanisms generating the time course of dual component excitatory synaptic currents recorded in hippocampal slices', *Neuron*, 5: 247-53.

Isaacson, J. S., and R. A. Nicoll. 1993. 'The uptake inhibitor L-trans-PDC enhances responses to glutamate but fails to alter the kinetics of excitatory synaptic currents in the hippocampus', *J Neurophysiol*, 70: 2187-91.

Kwon, H. B., and B. L. Sabatini. 2011. 'Glutamate induces de novo growth of functional spines in developing cortex', *Nature*, 474: 100-4.

Mattison, H. A., D. Popovkina, J. P. Kao, and S. M. Thompson. 2014. 'The role of glutamate in the morphological and physiological development of dendritic spines', *Eur J Neurosci*, 39: 1761-70.

O'Donovan, S. M., C. R. Sullivan, and R. E. McCullumsmith. 2017. 'The role of glutamate transporters in the pathophysiology of neuropsychiatric disorders', *NPJ Schizophr*, 3: 32.

Overstreet, L. S., G. A. Kinney, Y. B. Liu, D. Billups, and N. T. Slater. 1999. 'Glutamate transporters contribute to the time course of synaptic transmission in cerebellar granule cells', *J Neurosci*, 19: 9663-73.

Sarantis, M., L. Ballerini, B. Miller, R. A. Silver, M. Edwards, and D. Attwell. 1993. 'Glutamate uptake from the synaptic cleft does not shape the decay of the non-NMDA component of the synaptic current', *Neuron*, 11: 541-9.

Savasta, M., A. Dubois, and B. Scatton. 1986. 'Autoradiographic localization of D1 dopamine receptors in the rat brain with [3H]SCH 23390', *Brain Res*, 375: 291-301.

Scimemi, A., and M. Beato. 2009. 'Determining the neurotransmitter concentration profile at active synapses', *Mol Neurobiol*, 40: 289-306.

Scimemi, A., A. Fine, D. M. Kullmann, and D. A. Rusakov. 2004. 'NR2B-containing receptors mediate cross talk among hippocampal synapses', *J Neurosci*, 24: 4767-77.

Scimemi, A., H. Tian, and J. S. Diamond. 2009. 'Neuronal transporters regulate glutamate clearance, NMDA receptor activation, and synaptic plasticity in the hippocampus', *J Neurosci*, 29: 14581-95.

Wallis, J. D. 2007. 'Orbitofrontal cortex and its contribution to decision-making', *Annu Rev Neurosci*, 30: 31-56.

Wamsley, J. K., D. R. Gehlert, F. M. Filloux, and T. M. Dawson. 1989. 'Comparison of the distribution of D-1 and D2 dopamine receptors in the rat brain', *J Chem Neuroanat*, 2: 119-37.